EMBO
*reports*

# Chemokines and galectins form heterodimers to modulate inflammation

Veit Eckardt[1], Michelle C Miller[2], Xavier Blanchet[1], Rundan Duan[1], Julian Leberzammer[1], Johan Duchene[1] (ID), Oliver Soehnlein[1], Remco TA Megens[1], Anna-Kristin Ludwig[3], Aurelio Dregni[2], Alexander Faussner[1], Kanin Wichapong[4], Hans Ippel[4], Ingrid Dijkgraaf[4], Herbert Kaltner[3], Yvonne Döring[1], Kiril Bidzhekov[1], Tilman M Hackeng[4], Christian Weber[1,4,5,6], Hans-Joachim Gabius[3], Philipp von Hundelshausen[1,5,†,*] (ID) & Kevin H Mayo[2,†]

## Abstract

Chemokines and galectins are simultaneously upregulated and mediate leukocyte recruitment during inflammation. Until now, these effector molecules have been considered to function independently. Here, we tested the hypothesis that they form molecular hybrids. By systematically screening chemokines for their ability to bind galectin-1 and galectin-3, we identified several interacting pairs, such as CXCL12 and galectin-3. Based on NMR and MD studies of the CXCL12/galectin-3 heterodimer, we identified contact sites between CXCL12 β-strand 1 and Gal-3 F-face residues. Mutagenesis of galectin-3 residues involved in heterodimer formation resulted in reduced binding to CXCL12, enabling testing of functional activity comparatively. Galectin-3, but not its mutants, inhibited CXCL12-induced chemotaxis of leukocytes and their recruitment into the mouse peritoneum. Moreover, galectin-3 attenuated CXCL12-stimulated signaling via its receptor CXCR4 in a ternary complex with the chemokine and receptor, consistent with our structural model. This first report of heterodimerization between chemokines and galectins reveals a new type of interaction between inflammatory mediators that can underlie a novel immunoregulatory mechanism in inflammation. Thus, further exploration of the chemokine/galectin interactome is warranted.

**Keywords** chemotaxis; CXCL12; G protein-coupled receptor; galectin-3; lectin
**Subject Categories** Immunology; Signal Transduction; Structural Biology

## Introduction

Coordinated trafficking of leukocytes is central to host defense and inflammation. In order to regulate its timing and strategic course of action, various mediators (such as chemokines and adhesion/growth-regulatory galectins) are involved to orchestrate leukocyte recruitment [1–3]. The prototypic CXC chemokine CXCL12 plays a major role in many inflammatory and homeostatic situations. CXCL12 activates the Gi protein-coupled receptor CXCR4 that is expressed by hematopoietic cell types, including T cells, monocytes, and neutrophils, thus promoting their recruitment [4–6]. The CXCL12/CXCR4 axis plays a crucial role in the trafficking of these types of cells in immune homeostasis and in various acute and chronic inflammatory diseases, such as atherosclerosis and rheumatoid arthritis [7–11].

Structurally, chemokines consist of a three-stranded β-sheet and a C-terminal α-helix [12]. In solution, most chemokines form homodimers (CXCL12) or higher-order oligomers. Because of their structural homology, certain CXC and CC chemokines can form heterodimers with altered functionality compared to their homodimer counterparts [13,14]. For example, CXCL12 binds to CCL5 and inhibits its function, whereas CXCL4 enhances CCL5-mediated monocyte recruitment in atherosclerosis [15,16].

In addition to proteins, cell surface glycans convey signals relevant to pathophysiological processes. Their relatively complex and heterogeneous structures are read and translated by various tissue lectins to effect biological functions [17–19]. (β-)Ga(lactoside-binding) lectins (=galectins) are a class of potent cis/trans-acting modulators that function as bridging factors between their carbohydrate recognition domain (CRD) and cell surface glycoconjugates [20,21]. In particular, proto-type galectin-1 (Gal-1) and chimera-type galectin-3 (Gal-3) are involved in inflammatory cell recruitment

1 Faculty of Medicine, Institute for Cardiovascular Prevention, Ludwig-Maximilians-University, Munich, Germany
2 Department of Biochemistry, Molecular Biology & Biophysics, Health Sciences Center, University of Minnesota, Minneapolis, MN, USA
3 Faculty of Veterinary Medicine, Institute of Physiological Chemistry, Ludwig-Maximilians-University, Munich, Germany
4 Cardiovascular Research Institute Maastricht, Maastricht University, Maastricht, The Netherlands
5 German Centre for Cardiovascular Research, partner site Munich Heart Alliance, Munich, Germany
6 Munich Cluster for Systems Neurology (SyNergy), Munich, Germany
*Corresponding author. Tel: +49 89 4400 54353; E-mail: phundels@med.lmu.de
†These authors contributed equally to this work as senior authors

[22,23]. Both galectins are upregulated in inflammatory diseases such as atherosclerosis and osteoarthritis that also involve chemokines [24–27].

Structurally, all galectins share a highly conserved β-sandwich fold consisting of a six-stranded β-sheet on one face (S-face or sugar-binding face) and a five-stranded β-sheet on the opposing face (F-face) [28–30]. Gal-3 is unique among galectins, because it has a relatively long N-terminal tail (NT) extending out from its CRD. The NT is relevant to Gal-3 function, self-association, and serine phosphorylation, and it can be proteolytically truncated and fully cleaved from the CRD, thus explaining the term chimera type. Analogous to chemokines, galectins can also form homodimers and oligomers [31,32], as well as galectin/galectin heterodimers [33].

Until now, chemokines and galectins have been investigated as physically separate and functionally independent entities. Here, we test the hypothesis that they can associate as heterodimers with functional consequences, a hitherto unappreciated concept. We first demonstrate by screening that several CC and CXC chemokines can interact with Gal-3 and Gal-1, and then, we focus work on the specific case of CXCL12/Gal-3. Nuclear magnetic resonance (NMR) studies reveal that CXCL12 and Gal-3 form heterodimers and allow for a molecular dynamics (MD) simulation-based structural model to be made. This model is validated by investigating several Gal-3 CRD mutants (engineered by replacing key residues at the interface with CXCL12) that reveal reduced binding to CXCL12. Functionally, wild-type (WT) Gal-3 CRD, but not the mutants, blocks CXCL12-mediated leukocyte migration, indicating relevance of the structurally defined association. Impairing CXCR4 signaling by the CRD is presumably due to its capacity to build a ternary complex with the chemokine and its receptor on the cell surface.

## Results

### Physical interaction of Gal-3, Gal-3 CRD, and Gal-1 with CC and CXC chemokines

Our initial evidence supporting the new concept for interactions between chemokines and galectins was obtained by using a solid-phase immunoassay with membrane-adsorbed chemokines and biotinylated Gal-3 and Gal-1 in solution. Chemokine-dependent association in the mix was detected with horseradish peroxidase (HRP)-conjugated streptavidin (SA) and chemiluminescence. Figure 1A–D shows an exemplary image and qualitative analysis of a chemokine blot incubated with biotinylated Gal-3. Examination of a comprehensive panel of CC and CXC chemokines with these galectins revealed multiple cases of interaction with a similar binding pattern (Fig 1E). Figure EV1A–D shows an image with qualitative analysis of a chemokine blot incubated with biotinylated Gal-1.

Surface plasmon resonance (SPR) experiments were also performed with these chip-conjugated galectins and soluble chemokines. Excluding artifactual effects from surface adsorption, we found that results from both assays were consistent (Fig EV1E–G). A negative result from the solid-phase assay (e.g., CXCL9 and CXCL11 show no interaction with Gal-3) may be attributed to inactivation and/or inaccessibility of the binding site due to surface adsorption. SPR binding kinetics of CXCL12 with chip-immobilized

Gal-3 allowed us to derive a CXCL12/Gal-3 $K_D$ of 80 nM when Gal-3 was coupled via its sole cysteine (Fig 1F). Immobilized Gal-3 CRD bound CXCL12 with a slightly higher affinity of 34 nM (Fig 1G), suggesting that the NT of Gal-3 is not the site of interaction with CXCL12 and may even interfere with CXCL12 binding due to its transient interactions with the CRD [34]. The affinity of Gal-3 was about 10-fold higher than that of the proto-type (homodimeric) Gal-1, supporting the idea of a galectin-specific interaction (Fig 1H). Lactose, the canonical ligand for the galectin CRD, did not inhibit the interaction between CXCL12 and Gal-3 (Appendix Fig S1A and B). This result indicates that the contact region for the chemokine does not involve the canonical glycan-binding site on the S-face of the CRD, as is the case for pairing of Gal-3 CRD via the NWGR motif (W is central for lactose binding due to C-H/π-interaction) with Bcl-2 family proteins [35]. Besides, unfractionated heparin blocked the binding of Gal-3 CRD to CXCL12 (Appendix Fig S1C), indicating that CXCL12 residues relevant to glycosaminoglycan (GAG) binding contribute to the heterodimer interface.

We then compared the affinity of Gal-3 for CXCL12 with that for a panel of chemokines that showed a robust response to the Gal-3 chip (Figs EV1E and EV2A–F). Whereas CCL17 gave uninterpretable weak responses (Fig EV2C), the other chemokines examined (CCL1, CCL5 E66S, CCL22, CCL26, and CXCL11) gave well-detectable signals with $K_D$ values falling into the range from 7.9 to 99 nM (Fig EV2G). In this experiment, we used the E66S mutant of CCL5 that cannot form higher-order homooligomers, because WT CCL5 did not permit accurate determination of $K_D$ values. For subsequent structural and functional studies, we selected the Gal-3/CXCL12 pair due to its broad tissue distribution, co-expression in diverse organs, and biological relevance (Appendix Fig S2) [36].

In conclusion, Gal-3 interacts with CC and CXC chemokines primarily via its CRD, and without direct involvement of the canonical glycan-binding S-face of the CRD.

### Formation of CXCL12/Gal-3 heterodimers

To identify the interacting contact surfaces between CXCL12 and Gal-3, we performed $^1$H–$^{15}$N HSQC experiments with $^{15}$N-labeled CXCL12 and unlabeled Gal-3 CRD, as well as with $^{15}$N-labeled Gal-3 CRD and unlabeled CXCL12. Since CXCL12 itself forms relatively weak homodimers in fast exchange on the chemical shift timescale, the equilibrium between CXCL12 monomers and dimers can be shifted to mostly monomer by lowering the pH [37,38]. Addition of unlabeled Gal-3 CRD to $^{15}$N-labeled CXCL12 (and vice versa) resulted in some significant chemical shift changes as shown in the HSQC spectral expansions and chemical shift maps provided in Fig 2A and B. The entire HSQC spectra from which these expansions were made are provided in Appendix Fig S3A and B with some key interacting residues boxed in. Changes in resonance line widths and chemical shifts indicate that binding interactions occur in the intermediate exchange regime on the NMR chemical shift timescale, which in turn suggests that the heterodimer dissociation constant, $K_D$, falls in the $10^{-6}$ M range [39], slightly higher than from our SPR measurements. In addition, our HSQC data showed that lactose did not disturb the interaction, consistent with our SPR data (Appendix Fig S1A and B). Chemical shift changes identify regions of inter-protein contacts, as highlighted in orange and red on the structures of CXCL12 (Fig 2C) and Gal-3 CRD (Fig 2D). In support

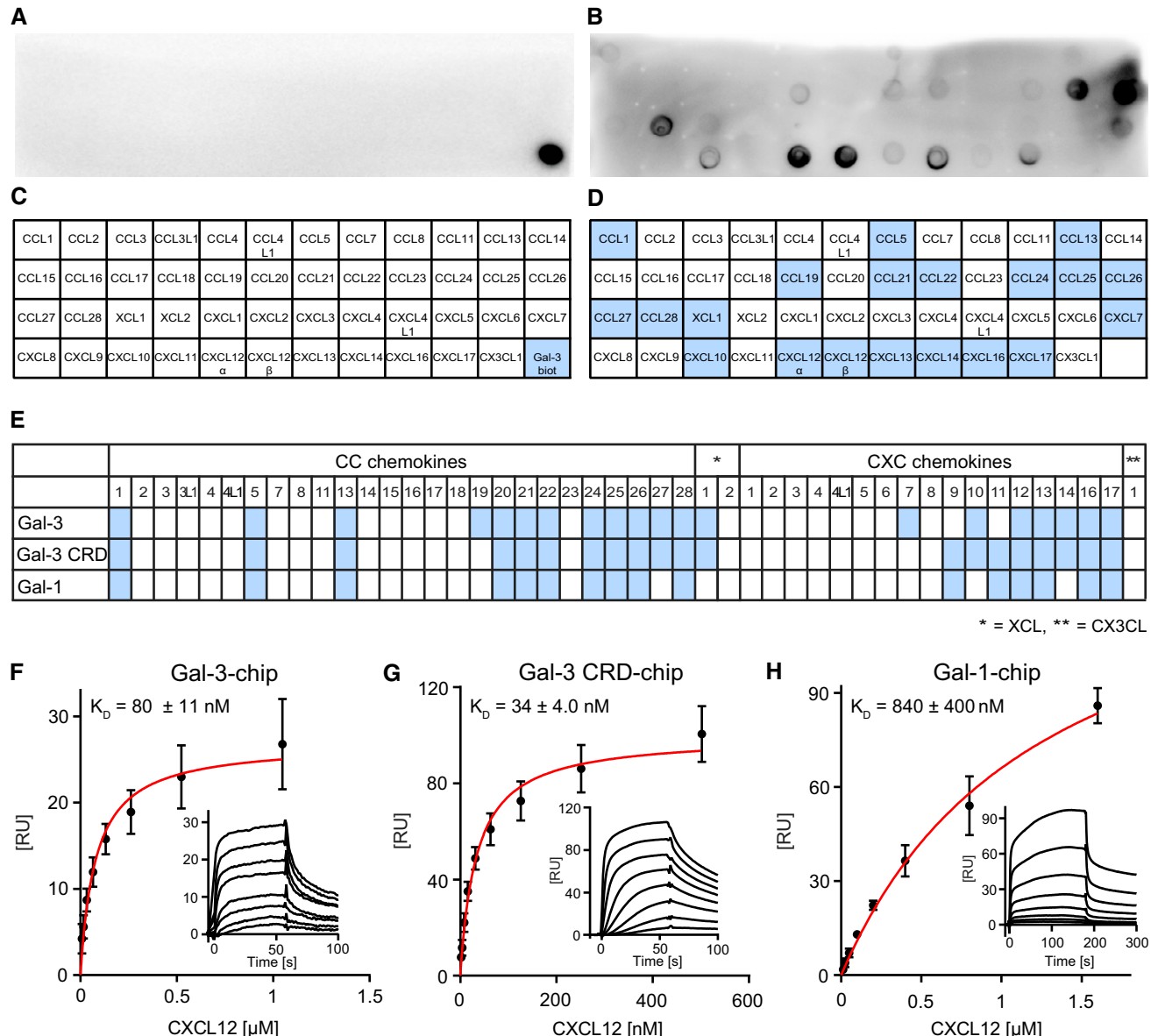

**Figure 1. Physical interaction of Gal-3 and Gal-1 with CC and CXC chemokines.**

A–E Chemokine–galectin interactions were detected by using a solid-phase immunoassay. For this, 46 human CC and CXC chemokines were adsorbed on nitrocellulose membranes and the stripes incubated in parallel with (A) TBS or (B) TBS containing biotinylated galectins (the representative image shows a processed membrane tested with labeled Gal-3). Signals had been generated by using SA-HRP and chemiluminescence reagents. (C, D) The blots were subjected to densitometric analysis, and (E) all independent experiments were combined (binding chemokines in light blue, Gal-3: $n = 5$, Gal-3 CRD: $n = 5$, Gal-1: $n = 4$).

F–H For SPR-based experiments, (F) Gal-3 (density 650 RU), (G) Gal-3 CRD (density 1180 RU), and (H) Gal-1 (density 130 RU) were immobilized and increasing concentrations of CXCL12 were passed over the flow cells. The red curve represents a single-site fit to the data. Insets are representative sensorgrams of CXCL12 testing on immobilized galectin. Data represent the mean ± SD from six (F) or three (G and H) independent experiments.

of these NMR data, silver staining of SDS–PAGE gels loaded with CXCL12, the cross-linker BS(PEG)$_5$, and increasing concentrations of Gal-3 CRD exhibited bands at the position expected for the CXCL12/Gal-3 CRD heterodimer (Appendix Fig S4).

Whereas Gal-3 CRD exists as a compact monomer in solution, full-length Gal-3 is characterized by intramolecular dynamics via transient backfolding of the NT onto the CRD F-face and a very weak tendency for self-association, both of which complicate structural interaction analyses [34]. Nevertheless, we assessed CXCL12-

induced chemical shift changes in $^{15}$N-labeled full-length Gal-3 and found that chemical shifts were overall reduced (Appendix Fig S5). In addition to the residues within the CRD, residues 5-15 near the N-terminus were also chemically shifted (Appendix Fig S5). This may reflect competition of chemokine/galectin heterodimer formation with intramolecular interactions between the NT and CRD [34]. In any event, our ligand blotting, SPR, and cross-linking results show that the NT of Gal-3 is not required for formation of Gal-3/CXCL12 heterodimers (Figs 1E and G, and EV1F, and Appendix Fig S4). Of

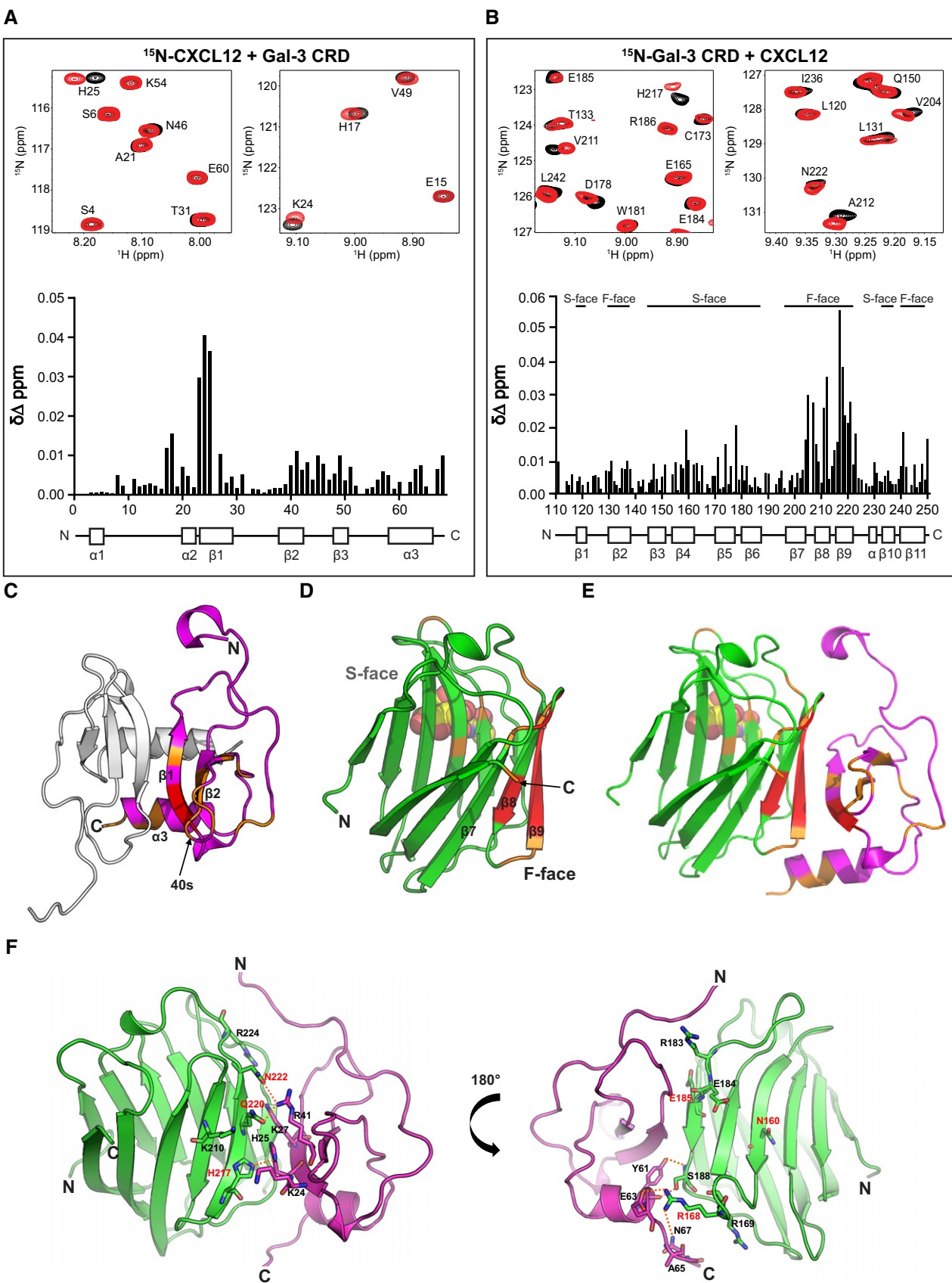

**Figure 2.**

**Figure 2. Formation of the CXCL12/Gal-3 CRD heterodimer.**

A, B  Expansions of $^1H$–$^{15}N$ HSQC spectra are overlaid, and Δδ values are plotted vs. the amino acid sequences, along with secondary structures: (A) 10 μM $^{15}N$-enriched CXCL12 alone (black peaks) and in the presence of 330 μM Gal-3 CRD (red peaks; assignments reported by Murphy et al [74]) and (B) 30 μM $^{15}N$-enriched Gal-3 CRD alone (black peaks) and in the presence of 500 μM CXCL12 (red peaks; assignments for Gal-3 reported by Ippel et al [34]).

C–E  X-ray crystal structures are depicted (C) for the CXCL12 homodimer (PDB access code 4UAI; monomer in magenta) and (D) for Gal-3 CRD bound with lactose (PDB access code 1A3K; in light green). (E) The energy-minimized structure of the CXCL12/Gal-3 CRD heterodimer was calculated by MD simulations. Residues with the largest Δδ values in HSQC spectra are highlighted in red (2SD-1SD) and orange (1SD-mean).

F  A model structure of the CXCL12/Gal-3 CRD heterodimer interface is shown with Gal-3 CRD in light green, the CXCL12 monomer in magenta, mutated residues in red, and hydrogen bonds as dashed orange lines.

note, analyses of interactions between Gal-1 and CXCL12 are likely hampered by homodimerization of Gal-1 that is favored over heterodimerization (data not shown), thus likely explaining its weaker affinity for CXCL12 compared to Gal-3 (Fig 1F and H).

Having identified mutual sites of interaction on CXCL12 and Gal-3 CRD (Fig 2A–D), we performed MD simulations started after manually docking the Gal-3 CRD to CXCL12 in 10 different rotational orientations that were all consistent with our HSQC data. Appendix Fig S6A shows the resulting heterodimer structures following energy minimization, with complex 6 being the energetically most favorable (−50 kcal/mol). Decomposition analyses of the free binding energies (ΔG) for each residue in the Gal-3 CRD (Appendix Fig S6B) and CXCL12 (Appendix Fig S6C) in complex 6 identify the residue pairing sites (Fig 2). Regions of contact in complex 6 (Fig 2E) correspond best to those identified by NMR (Fig 2A and B). A few specific residue pairings are depicted in Fig 2F. Several amino acids contribute to two major contact sites within the Gal-3 CRD at the strands β6 and β8-9 (CRD F-face), and the loop between β4 and β5. Gal-3 strands β6, β8, and β9 (in particular, residues E185 (β6), H217, Q220, and N222 (β9)) interact with a dominant binding region in the strands β1 and β2 of CXCL12 (in particular, R41 (β2), K27 (β1), and K24 (β1); please see Fig 2F, left panel). In addition, Gal-3 S188 (β6) and residues of the loop between β4 and β5 (i.e., R168 and R169) interact with the CXCL12 helix residues Y61, E63, A65, and N67 to establish a neighboring contact site (Fig 2F, right panel). These HSQC experiments thus revealed that CXCL12 engages in heterodimer formation with the Gal-3 CRD. Since the CRD F-face is located on the opposite side of the lactose-binding β-sheet S-face (Fig 2D), this explains why lactose does not reduce the binding of Gal-3 to CXCL12 and vice versa.

Using complex 6 (Fig 2E and F, Appendix Fig S6A) and decomposition analysis (Appendix Fig S6B and C), we selected several residues at the CXCL12/Gal-3 binding interface, mutated those residues in silico, and performed MD simulations to calculate BFE (Appendix Table S1). Whereas some Gal-3 mutants showed relatively small energetically favorable changes (e.g., K210D, Q220D, Q220E) in binding CXCL12, others showed highly unfavorable energetics (R168A, E185A, H217A, Q220A, Q220K, Q220R, N222A). Several positively charged residues in CXCL12 are located at the interface with Gal-3 (Fig 2F, left panel); Q220 of Gal-3 is one of them, such that introducing a negatively charged residue (i.e., glutamate) at that position (i.e., Q220E) might promote favorable electrostatic interactions and a more negative ΔG value as obtained in silico (Appendix Table S1). On the other hand, N160 lies on the opposing, non-interacting S-face in Gal-3 and is known to contribute to carbohydrate binding (Fig 2F, right panel).

For empirical validation of our model, we used site-directed mutagenesis to produce several mutants and assess effects on

heterodimer formation. Q220E in NMR and N160A in ligand blots and SPR were used as controls. Three Gal-3 CRD mutants (Q220E, Q220K, and H217A) were selected to assess their effects on HSQC spectra of $^{15}N$-labeled CXCL12 when examining mixtures. Even though all HSQC spectra look highly similar, analysis of the data could reveal distinct differences. Figure EV3 shows chemical shifts of $^{15}N$-labeled CXCL12 with each of these Gal-3 CRD mutants. These maps show the same trends as observed with WT Gal-3 CRD. Although this indicates that WT Gal-3 CRD and its mutants interact with CXCL12 in the same way, the magnitudes of Δδ changes are different. Compared to WT Gal-3, Q220E Δδ values are slightly increased (Fig EV3A), whereas those for Q220K and H217A are decreased (Fig EV3B and C). CXCL12 sequence-averaged Δδ values are 0.0061 ppm for WT Gal-3 CRD, 0.0073 for Q220E, 0.0036 for Q220K, and 0.0048 for H217A. Smaller chemical shift changes usually indicate weaker intermolecular interactions [39]. Here, average Δδ values suggest slightly stronger binding between CXCL12 and Q220E, and weaker binding between CXCL12 and Q220K and H217A. These trends parallel those observed in our MD-based free energy calculations, which yielded ΔG values of −50 kcal/mol for WT Gal-3 CRD, −58 kcal/mol for Q220E, −31 kcal/mol for Q220K, and −38 kcal/mol for H217A (Appendix Table S1).

Densitometric analysis of CXCL12 binding to variants of Gal-3 (Appendix Fig S7A) and Gal-3 CRD (Appendix Fig S7B) demonstrates that residues N222 and E185 are indeed involved in the interaction with CXCL12, whereas N160 is not. Similarly, the affinity of CXCL12 injected over sensor chips with immobilized Gal-3 mutants R168A, E185A, H217A, and Q220K was reduced, whereas the affinity of N160A was not (Appendix Fig S8A–H). In addition, Gal-3 mutant binding to the N-glycans of a common galectin binder, i.e., the glycoprotein asialofetuin (ASF), was only impaired in the case of N160A that showed no significant effect on heterodimer formation (Appendix Fig S8I–L), which supports our findings that CXCL12/Gal-3 heterodimer formation is not significantly affected by glycan binding (Appendix Fig S1A and B).

## Gal-3-mediated inhibition of CXCL12-induced leukocyte migration

We next investigated the functional consequences of CXCL12:Gal-3 heterodimerization. Initially, we examined whether Gal-3 affects CXCL12-induced migration of Jurkat T cells, and discovered that both Gal-3 and Gal-3 CRD inhibited chemotaxis in a dose-dependent manner (Fig 3A and B). In contrast, Gal-1 only inhibited migration at 1 μM (Fig 3C). These results with Jurkat cells were replicated using primary cells, i.e., activated human CD4$^+$ T cells (Fig 3D). A bell-shaped chemotaxis curve was observed upon increasing the

concentration of CXCL12, with the height of the curve being significantly reduced in the presence of 0.1 nM Gal-3 CRD (Fig 3 E and F). Consistent with our concept, the CXCL12 chemotaxis curve with Gal-1 remained unchanged even in the presence of 100 nM lectin (Fig 3G). Because the affinity of CXCL12 for Gal-1 is lower than that for Gal-3 and Gal-1 has no apparent effect on chemotaxis, we

assumed that chemotactic inhibition resulted from the physical interaction between Gal-3 and CXCL12. Galectins alone at 1 μM had no effect on Jurkat cell migration or viability (Fig 3H and I), consistent with a previous report [40].

Extending the scope of our investigation to other CXCR4-expressing cell types, we found that Gal-3 and Gal-3 CRD also inhibited

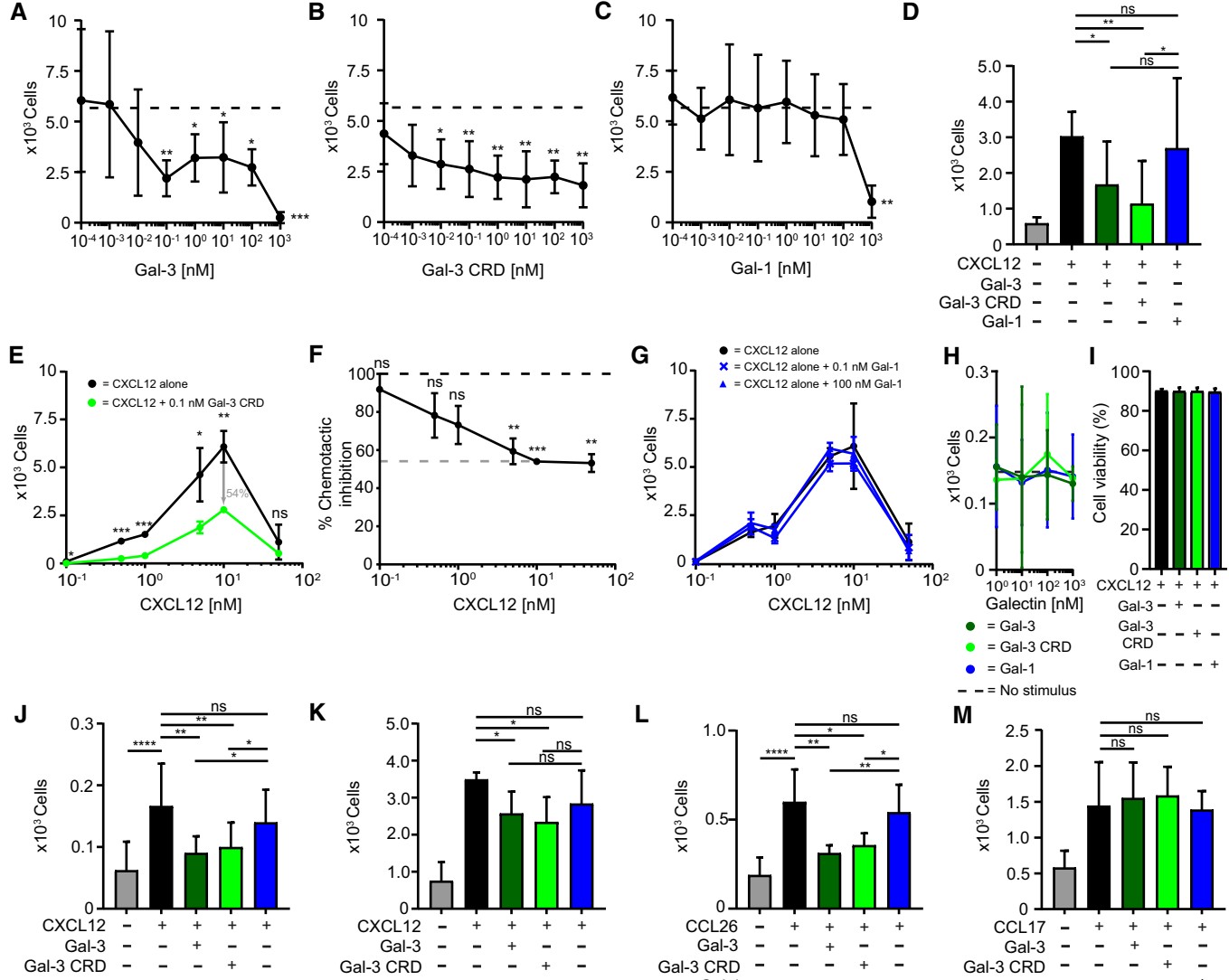

**Figure 3. Gal-3-mediated inhibition of CXCL12-induced leukocyte migration *in vitro*.**

A–C  Jurkat T cells migrated in the presence of 10 nM CXCL12 with increasing concentrations of (A) Gal-3, (B) Gal-3 CRD, and (C) Gal-1 (A–C: *n* = 4).

D  Human CD4+ T cells migrated in the presence of 10 nM CXCL12 alone and with 10 nM Gal-3 (dark green), Gal-3 CRD (light green), and Gal-1 (blue, *n* = 3).

E, F  (E) Jurkat T cells migrated to increasing concentrations of CXCL12 alone (black) and CXCL12 in the presence of 0.1 nM of Gal-3 CRD (light green). (F) The inhibitory effect of Gal-3 CRD is shown as percentage of the chemotactic effect of CXCL12 (E, F: *n* = 3).

G  Jurkat T cells were allowed to migrate in the presence of CXCL12 at increasing concentrations alone and with 0.1 nM Gal-1 or 100 nM Gal-1 (*n* = 3).

H  Jurkat T cells did not migrate in the sole presence of 1 nM to 1 μM Gal-3, Gal-3 CRD (both *n* = 5), or Gal-1 (*n* = 4).

I  The viability of Jurkat T cells was assessed after incubation with 1 μM CXCL12 alone and in the presence of Gal-3, Gal-3 CRD, and Gal-1 (*n* = 2).

J, K  (J) THP-1 cells and (K) neutrophils migrated in the presence of 10 nM CXCL12 alone and with 10 nM Gal-3, Gal-3 CRD, and Gal-1 (J, K: *n* = 3).

L  Human eosinophils migrated in the presence of 10 nM CCL26 alone and with 10 nM Gal-3, Gal-3 CRD, and Gal-1 (*n* = 3).

M  CD4+ T cells migrated in the presence of 1 nM CCL17 alone and with equimolar concentrations of Gal-3, Gal-3 CRD, and Gal-1 (*n* = 4).

Data information: Cell migration is shown as absolute cell counts. Data represent the mean ± SD from the indicated number of independent experiments each performed in three technical replicates, and these were statistically analyzed by using (A–C, F–G, J) a single sample *t*-test or (D, I, J–M) by an unpaired *t*-test against the effect of the chemokine alone or as indicated (*\*P* ≤ 0.05, \*\**P* ≤ 0.01, \*\*\**P* ≤ 0.001).

CXCL12-induced migration of monocytic THP-1 cells and neutrophils (Fig 3J and K), whereas Gal-1 caused only subtle effects. Supporting the idea of a functional impact from chemokine-Gal-3 interactions, we found that Gal-3 and Gal-3 CRD (both of which interact with CCL26) also inhibit CCL26-mediated migration of human eosinophils (Fig 3L). On the other hand, these galectins have no effect on CCL17-mediated migration of CD4$^+$ T cells (Fig 3M), consistent with our observation that Gal-3 and Gal-3 CRD do not interact with CCL17 (Figs 1E, and EV1E and F, and EV2C).

Under physiological conditions, co-injection of Gal-3 and CXCL12 into mice completely abrogated CXCL12-induced intraperitoneal (IP) recruitment of neutrophils (Fig 4A) and classical monocytes (Fig 4B) after 4 h. To find out whether genetic deletion of Gal-3 had an enhancing effect on IP recruitment of classical monocytes post-injection of thioglycolate (TG) broth into the peritoneum, responses in WT and KO mice were analyzed 18 h after stimulation with TG. As Fig 4C documents, (i) TG induces cell recruitment into the peritoneum, (ii) its extent is partially reduced by the CXCR4 antagonist to signal involvement of CXCR4-independent mechanisms, and (iii) Gal-3 absence increases recruitment, pointing to involvement of other chemokines as targets or of CXCR4-independent CXCL12 blocking. TG increased the amount of CXCL12 in the peritoneal lavage (Fig 4D) and reduced CXCR4 expression on classical monocytes (Fig 4E). The TG response was partly dependent on the presence of CXCR4, because pre-injection of the CXCR4 antagonist AMD3465 attenuated the effect (Fig 4C). Therefore, we surmise that these effects are attributable to the absence of CXCL12/Gal-3 interactions in these Gal-3$^{-/-}$ mice.

In addition, we performed antibody-based proximity ligation (PLA, Duolink®), demonstrating that CXCL12 and Gal-3 are in close proximity on cells recruited to the peritoneum after TG injection, and thus allowing for functional interactions under inflammatory conditions (Fig EV4A and B). To further substantiate the formation

of CXCL12/Gal-3 heterodimers *in vivo*, we stained Gal-3 and CXCL12 simultaneously in frozen sections of lymph nodes from WT and CXCL12$^{-/-}$ mice (Fig EV4C and D). Here, we found partial co-localization of Gal-3 and CXCL12 that was primarily detectable at the lymph node capsule where CXCL12-expressing lymphatic endothelial cells come into close proximity with subcapsular sinus macrophages (SSM). Further evidence for close contacts between CXCL12 and Gal-3 *in situ* was obtained by antibody-based PLA staining of lymph nodes extracted from WT and CXCL12$^{-/-}$ mice (Fig EV4E), indicating the potential for CXCL12 and Gal-3 to directly interact under physiological conditions.

To further make the case for this new type of pairing between inflammatory mediators, we found that the weaker interacting Gal-3 mutants E185A and N222A and Gal-3 CRD mutants E185A and N222A did not inhibit CXCL12-mediated chemotaxis up to 100 nM (Fig 5A and B). Similarly, CXCL12-mediated Jurkat cell migration was not inhibited by other weaker binding Gal-3 and Gal-3 CRD mutants, namely R168A, H217A, and Q220K (Fig 5C and D). In contrast, Gal-3 CRD mutant N160A, Gal-3 mutant Q220E, and Gal-3 CRD mutant Q220E (Fig EV3A) that did bind CXCL12 comparable to WT (Appendix Figs S7B, S8B, and Fig EV3A) did have an inhibitory effect (Fig 5C and D).

The observation that variants of Gal-3 at 1 μM inhibited chemotaxis independent of CXCL12, whereas Gal-3 CRD and its mutants E185A and N222A did not completely block cell migration (Fig 5A and B), may be explained by considering that the variants of full-length Gal-3 induce cell aggregation at 1 μM, an effect that was blocked by lactose (Fig 5E). We also performed transmigration assays with lactose (as well as with the disaccharide cellobiose that does not bind galectins) and with cells pre-treated with 1-deoxy-mannojirimycin hydrochloride (DMJ) that reduces the level of galectin-binding ligands on the cell surface by a shift to high-mannose-type N-glycans. As expected, the effect of Gal-3 and Gal-1

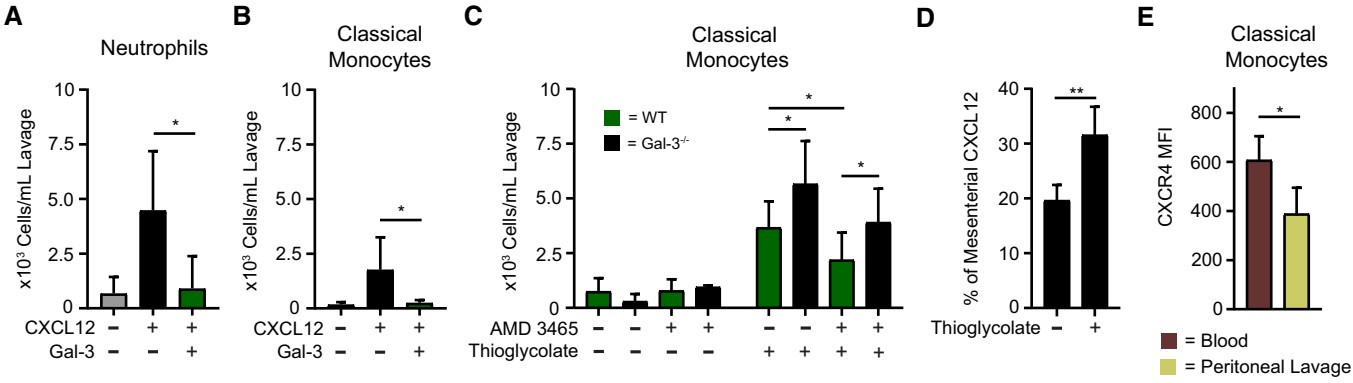

**Figure 4. Peritoneal recruitment of leukocytes by CXCL12 in the presence of Gal-3.**

A, B    The peritoneal recruitment of (A) CD45$^+$/CD115$^-$/Ly6G$^+$ neutrophils and (B) CD45$^+$/CD115$^+$/Ly6C$^{hi}$ classical monocytes in C57BL/6J mice was assessed 4 h after intraperitoneal (IP) injection of 500 nM CXCL12 alone (black) and in the presence of 50 nM Gal-3 (dark green; A, B: $n = 7$ mice).

C    The peritoneal recruitment of classical monocytes after IP injection of PBS ($n = 6$) or TG in C57BL/6J WT (dark green, $n = 10$ mice) and Gal-3$^{-/-}$ (black, $n = 5$ mice) mice was assessed after 18 h. Where indicated, the mice received an IP injection of CXCR4 antagonist AMD 3465 12 h prior to the experiment.

D    The concentration of CXCL12 concentration was determined by ELISA on the peritoneal lavage normalized with levels from the mesenterium ($n = 4$ mice).

E    CXCR4 expression levels on Ly6C$^{hi}$ monocytes of the blood and the peritoneal lavage after 18 h of TG stimulation were determined by flow cytometry and indicated as mean fluorescence intensity (MFI) ($n = 4$ mice).

Data information: Cell migration to the peritoneum is shown as cells/ml lavage. Data represent the mean ± SD from the indicated number of mice and were statistically analyzed by using the unpaired *t*-test, as indicated (*$P \leq 0.05$, **$P \leq 0.01$).

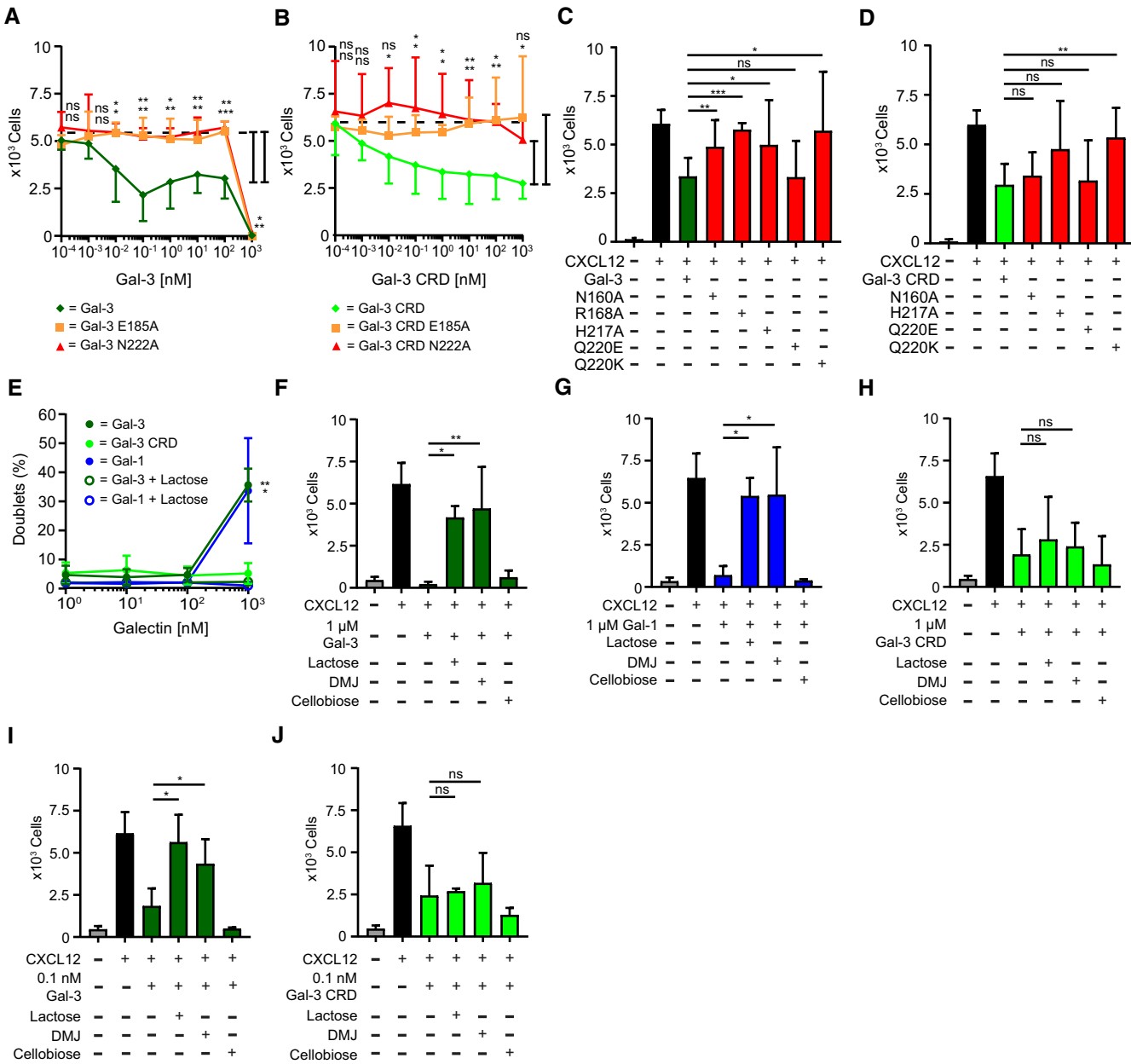

**Figure 5.   Inhibition of CXCL12-induced leukocyte migration by Gal-3 mutants and the role of galectin/glycan interactions *in vitro*.**

A, B   Jurkat T cells migrated in the presence of 10 nM CXCL12 with increasing concentrations of E185A (orange, *n* = 4) and N222A (red, *n* = 5) mutants of (A) Gal-3 and (B) Gal-3 CRD.

C, D   Jurkat T cells migrated in the presence of 10 nM CXCL12 with 10 nM (C) Gal-3 and (D) Gal-3 CRD mutants as indicated (C, D: *n* = 3).

E      Aggregation of Jurkat T cells in the presence of 1 nM to 1 μM Gal-3 and Gal-1 alone and with 70 mM lactose, and Gal-3 CRD, was determined by flow cytometry (*n* = 4).

F–J    Jurkat T cells migrated with 10 nM CXCL12 alone and in the presence of (F) 1 μM Gal-3, (G) Gal-1, (H) Gal-3 CRD, and (I) 0.1 nM Gal-3 and (J) Gal-3 CRD in the presence of 70 mM lactose or cellobiose or after treatment of cells with 150 μM DMJ.

Data information: Cell migration is shown as absolute cell counts. Data represent the mean ± SD from the indicated number of independent experiments each performed in three technical replicates, and these were statistically analyzed by using (A, B, E) a single sample *t*-test or (C, D, F–J) by an unpaired *t*-test against the effect of the chemokine alone or as indicated (*$P \leq 0.05$, **$P \leq 0.01$, ***$P \leq 0.001$).

at 1 μM was markedly reduced by the presence of lactose and DMJ (Fig 5F and G), whereas the effect of Gal-3 CRD was unaffected (Fig 5H). Because DMJ alone did not affect CXCL12-induced chemotaxis (Appendix Fig S9A), we assume that abrogation of chemotaxis

at 1 μM galectin concentration was caused by cell aggregation due to galectin oligomerization and cell–cell cross-linking.

Lactose and DMJ have no effect on CXCL12-mediated chemotaxis with Gal-1 (Appendix Fig S9B). However, they do reverse the

inhibitory effect of WT Gal-3 on CXCL12-mediated chemotaxis (Fig 5I). The reason they also do not have an effect with Gal-3 CRD (Fig 5J) is likely due to the presence of the NT in WT Gal-3, which complicates interpretation due to additional and unknown effects at the cell surface. In solution, the Gal-3 NT interacts transiently with the CRD F-face [34], a site of interaction that partially overlaps with the CXCL12 binding site on the lectin. Based on increased line broadening with WT Gal-3 in the presence of lactose, we know that lactose enhances NT binding to the CRD F-face, which results in attenuated CXCL12 binding to the lectin. Because DMJ treatment attenuates binding of galectins to glycans on the cell surface, it may be that some of them are necessary for optimal CXCL12/Gal-3 heterodimer formation and ensuing effects on the cell surface. Furthermore, we found that the small molecule CXCR4 agonist (NUCC-390) induces CXCL12-independent chemotaxis [41] that is unaffected by the presence of Gal-3 or Gal-3 CRD (Appendix Fig S9C).

For insight into the mechanism of Gal-3-mediated inhibition of CXCL12 function, we investigated effects of the galectin on CXCR4-mediated Gi signaling and β-arrestin 2 recruitment to CXCR4, a process that is relevant to chemotaxis [42].

### Inhibition of CXCL12-induced CXCR4 signaling by Gal-3 CRD

HEK 293 cells were transfected with CXCR4 and a luciferase-derived intracellular cAMP sensor. As expected, CXCL12 alone reduced cAMP levels reflecting Gi signaling, and Gal-3 CRD inhibited the effect from the chemokine over time (Fig 6A) and in a concentration-dependent manner (Fig 6B). In addition, we transfected HEK 293 cells with *Renilla* sp. luciferase II (RlucII)-conjugated CXCR4 and eYFP–β-arrestin 2 constructs to assess β-arrestin recruitment to the receptor by bioluminescence resonance energy transfer (BRET). CXCL12 caused recruitment of β-arrestin 2 that was prevented by Gal-3 CRD (Fig 6C). Unexpectedly, the effect of Gal-3 CRD was not accompanied by reduced internalization of CXCR4 (Fig 6D), which is mediated by β-arrestin recruitment [42]. However, it has been reported that chemokine receptors may signal in a biased fashion, with β-arrestin recruitment and internalization being uncoupled [42–44].

To confirm that Gal-3 CRD exerts its effect on CXCL12 via CXCR4, we performed cell-binding experiments with CXCL12 and Gal-3 using Jurkat T cells. First, we incubated the cells with CXCL12 and Gal-3, and demonstrated co-localization of the two proteins by an antibody-based PLA (Fig 6E). Next, we incubated the cells with fluorescently labeled Gal-3 CRD and unlabeled CXCL12 in the presence of AMD 3100, a competitive CXCR4 antagonist, and recorded fluorescence intensity by flow cytometry [45]. We found that the signal from the galectin in the presence of the chemokine was inhibited by AMD 3100 (Fig 6F). Furthermore, when we blocked direct binding of Gal-3 CRD to the cell surface with lactose, we observed an increase in the Gal-3 CRD signal upon addition of CXCL12. Once again, this effect was inhibited by AMD 3100 (Fig 6G). In contrast, unlabeled Gal-3 CRD did not displace fluorescently labeled CXCL12 from the cell surface (Fig 6H). Taken together, these findings suggest that Gal-3 CRD interacts with CXCL12, either having an indirect effect on CXCR4 via CXCL12 or directly binding to CXCL12 and CXCR4. Since glycan binding to the Gal-3 CRD is not required for the inhibition

of chemotaxis (Fig 5H and J), the involvement of an additional Gal-3 co-receptor is unlikely.

To test these hypotheses, we performed MD simulations of the CXCL12/Gal-3 CRD heterodimer interacting with CXCR4. Since the structure of CXCL12 bound to CXCR4 has so far not been determined, we superimposed complex 6 of the heterodimer (Fig 2E and F, Appendix Fig S6A) onto the structure of vMIP-II when associated with CXCR4 and removed the docked vMIP-II from the complex. The obtained model was then subjected to energy minimization in the course of a MD run over a period of 50 ns with coordinates and orientation from another monomer of CXCR4 (Fig EV5). The obtained structure illustrates that CXCL12's ligand property is not impaired by the Gal-3 CRD; that is, binding of the CXCL12/Gal-3 CRD heterodimer to CXCR4 is sterically possible. It may even be favored by direct interactions between Gal-3 CRD and CXCR4. This model obtained by MD simulation clearly warrants further investigation. Nonetheless, the experimental and computational lines of evidence converge to exclude galectin-dependent blocking of chemokine–receptor interaction for reducing CXCL12 activity as probed.

## Discussion

Chemokines and galectins regulate leukocyte recruitment and can be simultaneously upregulated under inflammatory conditions. In fact, in osteoarthritis, chemokines belong to a set of proteins that are upregulated in a NF-κB-dependent manner by Gal-3 [46]. Building on our discovery that CXC and CC chemokines form heterodimers with functional significance, we established a map of the chemokine interactome that illustrates numerous interactions [15]. Moreover, we recently reported on galectin/galectin heterodimer formation [33]. Due to the functional and structural similarities between chemokines and galectins, we hypothesized that members of both these effector molecule families may themselves interact to form chemokine/galectin heterodimers with functional consequences.

In the present study, we validated this hypothesis by demonstrating that Gal-1 and Gal-3 specifically interact with several chemokines in solid-phase immunoassays and SPR. When comparing the function of interacting and non-interacting chemokines, it is worthwhile to note that some chemokines primarily involved in later stages of inflammation, such as CCL22, CCL24, CCL26, and CXCL12, interact with the galectins, whereas chemokines, such as CCL2, CCL17, or CXCL8, that have been implicated in the initiation of inflammation [11,47–49] do not bind. Therefore, we propose a new concept that chemokine/galectin heterodimers may play a role in later stages of inflammatory processes or chronic inflammation.

Focusing on Gal-3 and CXCL12 that are both often found to be co-expressed and involved in inflammatory processes, we performed HSQC studies that revealed formation of a CXCL12/Gal-3 heterodimer in which Gal-3 binds to CXCL12 via the F-face of its CRD. The opposing glycan-binding S-face and the NT of Gal-3 are not part of the primary interaction domain. NMR analysis of the contact site between CXCL12 and Gal-1 or full-length Gal-3 was impeded by either intramolecular conformational changes or homodimerization. Viewed from the chemokine perspective, the interaction site includes the first β-strand (residues 17–27) and the

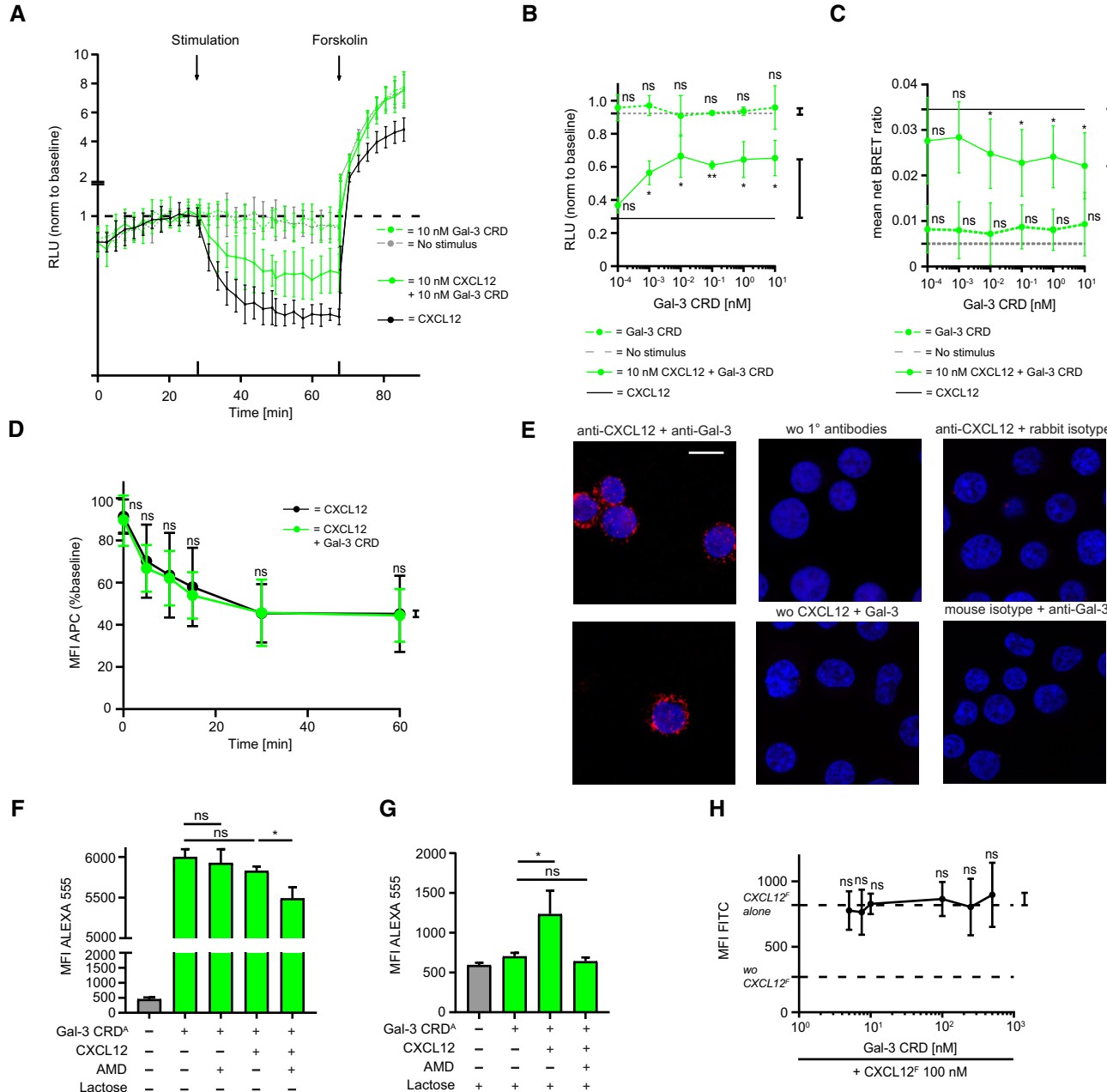

**Figure 6. Inhibition of CXCL12-induced CXCR4 signaling by Gal-3 CRD.**

A   HEK cells transfected with a luminescent cAMP sensor were incubated with 10 nM CXCL12 alone and in the presence of 10 nM Gal-3 CRD followed by stimulation with forskolin, an activator of adenylate cyclase. Results are shown as luminescence relative to baseline (RLU, example of *n* = 3).

B   The effect of 10 nM Gal-3 CRD upon stimulation with 10 nM CXCL12 prior to forskolin stimulation was tested. Control experiments were performed as indicated (*n* = 3, four technical replicates).

C   HEK cells transfected with a RlucII-conjugated CXCR4 and an eYFP–β-arrestin 2 construct were stimulated with 10 nM CXCL12 alone and in the presence of 10 nM Gal-3 CRD. Control experiments were performed as indicated. Results are given as the net BRET ratio (i.e., ratio of emissions at 535/485 nm minus the ratio of mock cells, *n* = 5, three technical replicates).

D   Internalization of CXCR4 with 10 nM CXCL12 alone and in the presence of 10 nM Gal-3 CRD was assessed by incubation with an APC-conjugated anti-CXCR4 antibody (baseline signal = 100%, *n* = 8, two technical replicates).

E   100 nM CXCL12 and Gal-3 were added to Jurkat T cells, and their co-localization was assessed on the cell surface by PLA. Control experiments were performed as indicated (representative example of *n* = 3). White scale bar: 10 μm.

F, G   Jurkat T cells were incubated with 100 nM of Gal-3 CRD Alexa Fluor 555 alone, with 10 μM AMD 3100, with 100 nM CXCL12, or with CXCL12 and AMD 3100. Experiments were performed (F) without and (G) in the presence of 70 mM lactose (F, G: *n* = 3, two technical replicates).

H   Jurkat T cells were incubated with 100 nM CXCL12 FITC alone and with increasing concentrations of Gal-3 CRD as indicated (*n* = 4, two technical replicates).

Data information: Data represent the mean ± SD of the indicated number of independent experiments and were statistically analyzed by (F, G) unpaired *t*-test or (B–D, H) single sample *t*-test against the effect of the chemokine alone or as indicated (*\*P* ≤ 0.05, *\*\*P* ≤ 0.01).

C-terminal α-helix (residues 59–68), as well as the second β-strand and 40s loop. Of note, these residues comprise the CXCL12 homodimer interface in the first β-strand and the C-terminal α-helix [50]. Although acidic conditions shift the monomer–dimer equilibrium toward monomer, Gal-3 may interfere with CXCL12 homodimerization and thus affect CXCR4 signaling under more physiological conditions [38,51,52]. Of particular note, the CXCL12/Gal-3 interface contains the GAG-binding motif of CXCL12, which primarily involves the first β-strand and 40s residues. Implying physiological significance, this region is reported as being required for chemokine presentation by endothelial cell GAGs [52,53]. Leading to an anti-chemokine effect, Gal-3 may specifically block this binding, a scenario that may impact on chemokine activity, as an antibody does [54]. The finding that heparin prevents CXCL12 from binding to Gal-3 CRD points into this direction so that interactions of CXCL12 with cognate GAGs and Gal-3 CRD are mutually exclusive. Studying the effect of Gal-3 on CXCL12 activity in a GAG-deficient cell line will be an approach to contribute to answer the arising question on the role of GAGs *in situ* for reducing CXCL12 activity by the galectin.

Here, we have provided initial support for our hypothesis that heterodimer formation affects chemokine function by performing transmigration assays and using a murine model of peritonitis. Whereas Gal-3 inhibited CXCL12-induced chemotaxis, Gal-1 (that exhibited markedly reduced affinity in SPR), the non-interacting chemokine CCL17 and the interaction-deficient mutants Gal-3 R168A, Gal-3 E185A, Gal-3 H217A, Gal-3 Q220K, and Gal-3 N222A had a reduced effect, if any. At this moment, it is puzzling that the effective concentrations of Gal-3 seem unexpectedly low and let it appear that a substantial portion of CXCL12 may not be active. Concerning the first point, should the chemokine be arranged in clusters at high density (presenting multivalency to the lectin), then a gradient of fractional affinity constants could arise with very high affinity for the first step, as documented for Gal-3 CRD association with a nonavalent glycoprotein [55]. Galectins bind to the multivalent glycoprotein asialofetuin with enhanced affinities and a gradient of decreasing binding constants [55]. With respect to low effective chemokine concentrations, although not yet mechanistically fully understood, it appears reasonable to postulate the following: that competition between galectins and GAGs (these glycan chains are discussed as direct or indirect factors for affecting chemokine availability [56]) for the chemokine may have a tangible bearing on CXCL12 activity, as also noted above.

On the cellular level, we showed that Gal-3 CRD affects CXCR4 signaling without interfering with receptor internalization. These results, together with the computational modeling of the ternary complex, suggest that the CXCL12/Gal-3 heterodimer binds CXCR4 rather than preventing the chemokine from interacting with its receptor. The possibility that CXCL12, when associated with the Gal-3 CRD, "may not be as potent as CXCL12 (alone) in triggering intracellular signals" (as suggested by a reviewer) can establish a mechanism toward galectin-mediated reduction of chemokine activity.

The biological functions of extracellular Gal-3 that have so far been described mostly depend on the glycan-binding capacity of its CRD and the formation of aggregates (lattice) with counterreceptors involving the CRD and possibly its NT [57,58]. To give an example from immune regulation, Gal-3 impedes diffusion of the glycosylated cytokine interferon-γ by cross-linking the cytokine with components of the extracellular matrix in a glycan-dependent manner [59]. Thus, Gal-3 can inhibit chemokine effects through both direct and indirect mechanisms. Moreover, Gal-3 induces neutrophil expression of CXCL8 in a glycan-dependent manner. Cleavage of its NT by neutrophil elastase renders the resulting Gal-3 CRD non-functional [60]. Our study has broadened this functional profile by demonstrating that regions of extracellular Gal-3 that are not involved in carbohydrate binding may in fact modulate inflammation. Our observations suggest that the Gal-3 CRD may have an anti-inflammatory role that could be exploited therapeutically. These findings do not exclude S-face-dependent effects of the galectin CRD in a physiological setting. Whether both sites operate simultaneously should prompt further studies. Since homodimeric Gal-3 variants have recently become available [61], this protein and also Gal-1/3 heterodimers offer perspectives on how to resolve this issue and may inspire biomedical efforts.

Since it was shown that CXCL12 and Gal-3 are simultaneously upregulated under chronic inflammatory conditions, proinflammatory Gal-3 may specifically block excessive or persistent inflammation by interfering with CXCL12 activity. Proteolytic Gal-3 truncation to its CRD by matrix metalloproteinases may be a control mechanism that attenuates or resolves inflammation. This assumption inspires the idea that a multimeric Gal-3 CRD construct (with active CRD or a mutant) may efficiently block CXCL12 activity *in situ*, while Gal-1 oligomerization appears to increase aspects of its biomedical activity [62,63]. The interaction of a lectin with a non-glycan counterreceptor, together with such engineering, can thus have a biomedical potential [64]. As a promising target, endothelial cell-derived CXCL12 drives atherosclerosis which underlies coronary heart disease [65,66]. Considering our discovery that Gal-3 is a potential antagonist of CXCL12, the present evidence for chemokine/galectin heterodimerization will prompt further investigation into the inter-family interactome and its pathophysiological relevance. Since Gal-3 fulfills criteria of an alarmin (or damage-associated molecular pattern) and a mediator of autophagy [67,68], the detection of CXCL12 binding will also warrant to explore its capacity to engage in pairing in respective processes.

# Materials and Methods

### Galectins, chemokines, and asialofetuin

#### Galectins

Gal-1 and Gal-3 were produced using *Escherichia coli* BL21(DE3) pLysS cells and the pGEMEX-1 expression vector (Promega). Cells were cultured at 37°C until an optical density$_{600}$ value of 0.6–0.8 was reached; then, protein production was induced by 100 μM of IPTG (isopropyl-β-D-thiogalactoside) and cultivation continued at 37°C (Gal-1) or 22°C (Gal-3) for 16 h. Proteins were purified from extracts by affinity chromatography on lactosylated Sepharose 4B as crucial step, and lactose was removed by gel filtration [69]. Isotopic labeling of full-length Gal-3 (residues 1–250) for NMR-spectroscopical analysis was done by using [15N]NH$_4$Cl as medium additive for production at 30°C for 16 h in the presence of 100 μM IPTG [34]. The Gal-3 CRD (residues 108–250) was generated by on-bead collagenase treatment (1 mg/10 mg of protein) for 16 h at 4°C [70].

One-site Gal-3 mutants (N160A, E185A, and N222A) were obtained by mutagenesis of the pGEMEX-1-Gal-3 and pGEX-6P-2-Gal-3 vectors using the QuikChange Mutagenesis Kit (Agilent Technologies, Waldbronn, Germany). All proteins were routinely checked for purity by one- and two-dimensional gel electrophoresis under denaturing conditions and for activity by solid-phase/cell-binding assays.

### Chemokines

For the array of the solid-phase immunoassay, chemokines were purchased from PeproTech (Rocky Hill, NJ, USA). WT human CXCL12α was bacterially expressed using a codon-optimized cDNA (Genscript, Piscataway, NJ, USA) as a thioredoxin-His-tagged fusion protein from the pET-32(+) vector with an enterokinase cleavage site at the N-terminus. The plasmid was transformed into *E. coli* BL21(DE3) cells and grown at 37°C in either Luria–Bertani or $^{15}$N-enriched Spectra 9 medium (Cambridge Isotope Laboratories, MA, USA). CXCL12α was purified from inclusion bodies. After separation using a HisTrap HP column (GE Healthcare, Chicago, IL, USA), the sample was dialyzed against 50 mM Tris (tris(hydroxymethyl) aminomethane) buffer (pH 8), filtered, and loaded on a Heparin HP column. The bound protein was eluted in 50 mM Tris/2 M NaCl (pH 8) and further dialyzed against 50 mM Tris/2 mM cysteine (pH 8) before cleavage using Enterokinase (Novagen, Merck, Darmstadt, Germany). The cleaved protein was purified using a Mono S 5/50 GL column (GE Healthcare). The fractions containing the protein were pooled, dialyzed against 1% acetic acid, lyophilized, and stored at −20°C until further use. The correct mass of CXCL12α was confirmed by mass spectrometry.

### Conjugation of proteins with biotin and fluorescent labels

Fluorescent and biotinylated proteins were prepared with the succinimidyl ester of Alexa Fluor 555 and the N-hydroxysuccinimide ester derivatives of FITC and biotin (all from Thermo Fisher Scientific, Waltham, MA, USA) according to the manufacturer's protocol. The conjugate was separated from the reagent using Sephadex G25 in PD-10 Desalting Columns (GE Healthcare). Preservation of activity was checked by cell signaling, hemagglutination, and binding assays using (neo)glycoproteins as matrix.

### Asialofetuin

Desialylation of fetuin from fetal calf serum (Sigma-Aldrich, Taufkirchen, Germany) was performed by hydrolysis in 0.05 N sulfuric acid at 80°C for 1 h, and the product (ASF) was purified by fast protein liquid chromatography on a Superdex 75 column (GE Healthcare).

## Cells, cell culture, and cell transfection

*Jurkat T cells* (clone E6-1, ATCC) and *human monocytic THP-1 cells* (no. ACC-16, DSMZ) were cultured as recommended by the supplier.

### Human CD4$^+$ T cells

PBMCs were separated from whole blood by density gradient centrifugation. CD4$^+$ T cells were isolated from PBMCs with the Dynabeads Untouched Human CD4$^+$ T Cells Kit, stimulated with the Dynabeads Human T-Activator CD3/CD28 Kit (both from Invitrogen, Thermo Fisher Scientific) for 3 days, and expanded in the presence of the Dynabeads with 30 U/ml human IL-2 for another 3 days.

### Human eosinophils

Whole-blood components were separated by density gradient centrifugation. The erythrocyte pellet was lysed, and eosinophils were isolated with the Eosinophil Isolation Kit in an unlabeled manner (Miltenyi, Bergisch Gladbach, Germany).

### Human neutrophils

Neutrophils were separated from whole blood by density gradient centrifugation.

### HEK 293 cell transfection for Gi signaling

The sequence of the luciferase-cAMP binding site fusion protein from the pGloSensor-20F vector (Promega) was amplified and ligated into a bicistronic pIRESneo vector (Clontech, Mountain View, CA, USA) to obtain the reporter gene plasmid. The pcDNA5/FRT/TO vector (Invitrogen) was used to express the CXCR4 receptor constructs (cDNA Resource Center, Bloomsburg University, Bloomsburg, PA, USA) [71]. Flp-In T-REx 293 cells (HEK 293, Invitrogen) were first stably transfected with the reporter plasmid using the Flp-In system (Invitrogen) and EcoTransfect (OZ Biosciences, Marseille, France). Stable clones were selected with 1 mg/ml geneticin. A suitable clone was then chosen as host cell line for stable overexpression of the CXCR4 construct using the Flp-In system with 250 µg/ml hygromycin B for selection.

### HEK 293 cell transfection for β-arrestin 2 recruitment

HEK 293 cell monolayers at 90% confluency on a 24-well plate were transiently transfected with 0.05 µg/well CXCR4-RlucII construct (Promega) and 0.2 µg/well eYFP–β-arrestin 2 construct or mock plasmid with 1 µl EcoTransfect. After 24 h, the cells were transferred to a black 96-well plate.

## Solid-phase immunoassays

100 ng samples of human chemokines or galectins were blotted onto a nitrocellulose membrane, and incubated with 200 nM biotinylated galectins or 120 nM CXCL12 overnight, signals developed with SA-HRP for 1 h, and added enhanced chemiluminescence substrate (Thermo Fisher Scientific). Densitometric analysis of digital records was performed using the program ImageJ.

## Cross-linking protein interaction analysis

Galectins were incubated with 1 mM BS3 or BS(PEG)$_5$ cross-linkers (both from Thermo Fisher Scientific) in 20 mM HEPES (pH 8.3) at room temperature. After 10 min, CXCL12 was added and the mixture further incubated for 1 h. The reaction was then stopped by addition of 1 M Tris (pH 8). Samples were analyzed by SDS–PAGE followed by silver staining according to the manufacturer's instructions (Alphalyse, Odense, Denmark).

## Surface plasmon resonance measurements

SPR experiments were performed on a Biacore X100 system (GE Healthcare).

## Binding experiments

Biotinylated Gal-3 and Gal-1 were immobilized on SA sensor chips at densities specified in the figure legends (Fig EV1E and G). Gal-3 CRD was immobilized using thiol-coupling chemistry on a NeutrA-vidin (Thermo Fisher Scientific)-modified C1 sensor chip at the specified density (Fig EV1F). 100 nM chemokines dissolved in HEPES-buffered saline with EDTA and surfactant P20 (HBS-EP+) were perfused at a flow rate of 30 µl/min for 3 min. The response in resonance units (RU) was recorded 20 s after the end of the injection.

## Kinetics experiments

Biotinylated (Gal-1 and Gal-3) and biotin-free (Gal-3, Gal-3 CRD, and mutants) galectins and biotinylated asialofetuin were immobilized on NeutrAvidin or using thiol- or amine-coupling chemistry (Thermo Fisher Scientific) on C1 or SA sensor chips at surface densities specified in the figure legends (Figs 1F–H and EV2A–F, and Appendix Figs S1A–C and S8A–L). CXCL12 and Gal-3 in HBS-EP+ were perfused for 1 min followed by a dissociation phase of 3 and 2 min for the chips with conjugated galectins and the chip presenting the glycoprotein ASF, respectively.

## Nuclear magnetic resonance spectroscopy

NMR samples were prepared in 3-mm NMR tubes. Typically, chemokine and galectin samples were buffer-exchanged and concentrated into 20 mM sodium acetate buffer, pH 4.5, and 4.5 mM lactose through five ultracentrifugation steps over Amicon Ultra-4 3-kDa filter devices (Merck, Darmstadt, Germany). Mixtures of CXCL12 and Gal-3 CRD, Gal-3, or Gal-1 at defined molar ratios were prepared from these stock solutions, and 5% (v/v) $D_2O$ was added for field locking, together with a trace of DSS as an internal chemical shift standard. $^1H$–$^{15}N$ HSQC experiments with a flip-back pulse and decoupling in the presence of scalar interactions, and nuclear Overhauser effect were recorded at 37°C on Bruker Avance III HD 700- and 850-MHz spectrometers equipped with cryogenically cooled triple resonance inverse probes. Spectra were processed and analyzed using Bruker TopSpin 3.2 and Sparky 3.114 software (T. D. Goddard, D. G. Kneller, SPARKY 3, the University of California, San Francisco, CA, USA). Resonance assignments of CXCL12 and Gal-3 CRD were performed by 2D NOESY and 3D-edited NOESY spectra. Chemical shift differences ($\Delta\delta$) induced upon binding were calculated as follows: $[(\Delta^1H)^2]^{1/2} + [(0.25\Delta^{15}N)^2]^{1/2}$ (in $^1H$ ppm). $\Delta$Intensity was calculated as follows: $1 - Int_i/Int_0$, where $Int_i$ is the resonance intensity of resonances of CXCL12 or Gal-3 CRD in the presence of the other component, respectively, and $Int_0$ is the intensity of CXCL12 or Gal-3 CRD resonances in its absence. The same experimental procedure was used for Gal-1 and Gal-3 proteins, as well as for Gal-3 mutants.

## Molecular dynamics simulations

The CXCL12/Gal-3 CRD heterodimer was subjected to MD simulations for 50 ns as described except applying Amber 14SB force field with TIP3P water models using Amber16 [72]. MD simulations were performed for 50 ns. Snapshots between 40 and 50 ns were extracted for binding free energy calculation using the MM/GBSA approach, and the BFE values were approximated from enthalpy values as described [72]. Default parameters were applied for BFE calculation, except using the generalized Born model 8 to compute the free energy of solvation. Ternary complex modeling between CXCL12, Gal-3 CRD, and CXCR4 was performed as described above. PDB access codes are as follows: CXCL12 homodimer (4UAI), Gal-3 CRD (1A3K), vMIP-II bound to CXCR4 (4RWS), and CXCR4 homo-dimer (3ODU).

## Chemotaxis

### Transwell migration

Chemotaxis assays were performed in triplicate with the number of independent experiments as stated in each respective figure legend. Chemokines and galectins in 230 µl of RPMI 1640/0.5% BSA were pipetted into the bottom well of a Transwell-96 permeable support (Corning, NY, USA) with 3.0 µm pore size for human granulocytes and 5.0 µm for all other cell types. $10^5$ cells in 70 µl were pipetted on top of the filter and allowed to migrate for 2 h for primary cells and 4 h for Jurkat and THP-1 cells. 20 µl of a 0.05 M EDTA solution was added to the bottom well, and the plates were incubated for another 15 min. The number of cells at the bottom of a well was measured by flow cytometry of the cell suspension for 30 s at medium speed (FACSCanto II, BD Biosciences, Franklin Lakes, NJ, USA).

## Cell aggregation

Jurkat T cells were incubated with the indicated concentrations of galectins for 4 h and subsequently analyzed by flow cytometry (FACSCANTO II). The forward scatter area (FSC-A) was plotted against the forward scatter width (FSC-W) to discriminate between singlets (low FSC-A, low FSC-W) and doublets/multiplets (high FSC-A, high FSC-W).

## Apoptosis

Cell viability was determined by fluorescence-activated cell scanning with the FITC Annexin V Apoptosis Detection Kit with 7-AAD (7-amino-actinomycin D) (BioLegend, San Diego, CA, USA).

## Mice

All animal experimental procedures were designed and conducted in agreement with the German Animal Welfare Legislation, and were reviewed and approved by local authorities (Regierung von Oberbayern, Munich, Germany). All mice were housed in IVC units and maintained on a 12-h dark/12-h light cycle. C57BL/6J mice were from Janvier (Le Genest-Saint-Isle, France), and tamoxifen-inducible general Cre-deleter C57BL/6-Gt(ROSA)26Sor$^{tm9(Cre/ESR1)Arte}$ (CreERT2) mice were from Taconic (TaconicArtemis GmbH, Cologne, Germany) and crossed with CXCL12$^{flox/flox}$ mice (on the Apoe$^{-/-}$ background), which were generated as described [15]. CXCL12$^{-/-}$ and WT mice were littermate offsprings from CreERT2$^+$, CXCL12$^{flox/flox}$, or CreERT2$^+$CXC12$^{wt/wt}$ mice after application of tamoxifen. Mice deficient for Lgals3 (coding for Gal-3) were from EUCOMM (C57BL/6N-Lgals3tm1a(EUCOMM)Wtsi/H, Strain ID EM:06800) [73].

### Murine model of peritonitis

500 nM CXCL12 alone, or in combination with 50 nM Gal-3, in PBS was injected into the peritoneal cavity of C57BL/6J mice (Janvier). To ensure efficacy in this physiological system, the chemokine/galectin molar ratio (10:1) was used. Mice were euthanized after 4 h. HBSS/0.3 mM EDTA/0.06% BSA was first injected into the peritoneal cavity, and then collected.

Peritonitis was induced by IP injection of 0.5 ml of 4% sterile thioglycolate broth. C57BL/6J mice were pre-treated IP with 125 μg of the CXCR4 antagonist AMD3465 (Tocris Bioscience, Bristol, UK) 12 h earlier. The peritoneal lavage was obtained after 18 h.

Cells were stained with a mixture of fluorescent antibodies and analyzed by flow cytometry. Only single cells (FSC-H/FSC-W$^{low}$) were gated. B cells (B220$^+$) and macrophages (F4/80$^+$) were excluded, and classical monocytes (CD115$^+$/Ly6C$^{hi}$) and neutrophils (CD115$^-$/Ly6G$^+$) were gated from CD45$^+$ (Appendix Fig S10A). Percentages of leukocyte subsets of untreated and TG-treated mice are indicated (Appendix Fig S10B–H). All antibodies were obtained from eBioscience (Thermo Fisher Scientific).

### Proximity ligation assay

Lymph nodes for frozen sections were explanted from C57BL/6 mice. 10$^6$ Jurkat T cells were incubated with CXCL12 and Gal-3 at 4°C for 1 h, fixed, and mounted onto poly-ʟ-lysine (Sigma-Aldrich)-coated slides. On all samples, sites for non-specific protein binding were blocked and the cells or sections were incubated with 5 μg/ml of a polyclonal goat anti-mouse CXCL12 (Bio-Rad Laboratories, Hercules, CA, USA) and 2.5 μg/ml of a rabbit anti-human Gal-3 (affinity-purified IgG) antibody at 4°C overnight. Samples were incubated with secondary antibodies conjugated to complementary oligonucleotides that were ligated and amplified according to the manufacturer's instructions (all reagents from the Duolink In Situ Red Goat/Rabbit Kit; Sigma-Aldrich). Photomicrographs were taken using a confocal microscope (SP8; Leica, Wetzlar, Germany; magnification × 100, numerical aperture 1.4, oil immersion) and processed with LAS X (Leica) and Huygens software.

### CXCR4 signaling

#### Gi signaling

HEK 293 cells expressing the GloSensor (Promega, Madison, WI, USA) and CXCR4 were cultured in a black 96-well plate (Perkin Elmer, Waltham, MA, USA) for 2–3 days until the cells were confluent. Cells were then incubated with a HBSS/20 mM HEPES/2.5% Luciferin-EF (Promega) solution for 2 h. Luminescence was determined using a plate reader (infinite F2000PRO; Tecan, Männedorf, Switzerland) until steady state was achieved. Forskolin (1 μM) was added 28 min after the stimulus, and the luminescence was recorded.

#### β-Arrestin 2 recruitment

HEK 293 cells expressing eYFP-β-arrestin 2 and CXCR4 *Renilla* sp. luciferase II (RlucII) were cultured to confluence in a black 96-well plate with 0.5 μg/ml tetracycline used to induce expression. Total fluorescence was determined at 535 nm. Coelenterazine (15 μM) was added, and total luminescence was detected at 485 nm. The stimulus was added, and the BRET ratio (emissions at 535 nm/485 nm) was determined.

### Internalization of CXCR4

10$^5$ Jurkat T cells were incubated with CXCL12 and Gal-3 CRD at 37°C, fixed in 4% PFA, and stained with a monoclonal (12G5) APC-conjugated anti-human CXCR4 antibody (BD Biosciences).

### Galectin and CXCL12 binding to T cells

10$^5$ Jurkat T cells were incubated as stated in the figure legend (Fig 6F–H) for 1 h. Where indicated, cells were pre-incubated with AMD 3100 (Sigma-Aldrich) for 15 min, and experiments were performed in the presence of the compound. Fluorescence signals were recorded by flow cytometry.

**Expanded View** for this article is available online.

### Acknowledgements

This work was supported by funds from the National Science Foundation (BIR-961477), the University of Minnesota Medical School, and the Minnesota Medical Foundation (K.H.M) and by the Deutsche Forschungsgemeinschaft [SFB914, B08 (O.S. and C.W.), SFB1123, A1 (C.W. and Y.D.), A2 (P.v.H and H.-J.G.), and Z1 (R.T.A.M), INST 409/150-1 FUGG (C.W. and R.T.A.M)]. At Maastricht University, C.W. is Van de Laar professor of atherosclerosis, and K.H.M. is Van de Laar professor of structural biology. K.H.M. also gratefully acknowledges support for a visiting professor fellowship at the Ludwigs-Maximilian-Universität (LMU) from the LMU Center of Advanced Studies, as well as from the Alexander von Humboldt-Stiftung. We are all most grateful to the reviewers for their detailed, expert input during the review process.

### Author contributions

VE designed and performed experiments, analyzed and interpreted results, and wrote the manuscript; MCM, XB, RD, JL, JD, OS, RTAM, A-KL, AD, AF, KW, HI, ID, HK, YD, and KB designed and performed experiments; TMH, H-JG, and CW supervised the study and made critical revisions to the manuscript; PvH and KHM conceived the study, designed experiments, analyzed and interpreted results, wrote the manuscript, and share senior authorship.

### Conflict of interest

The authors declare that they have no conflict of interest.

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
