## [Review Process File · EMBO Reports]

Chemokines and galectins form heterodimers to modulate inflammation

Veit Eckardt, Michelle C. Miller, Xavier Blanchet, Rundan Duan, Julian Leberzammer, Johan Duchene, Oliver Soehnlein, Remco T.A. Megens, Anna-Kristin Ludwig, Aurelio Dregni, Alexander Faussner, Kanin Wichapong, Hans Ippel, Ingrid Dijkgraaf, Herbert Kaltner, Yvonne Doring, Kiril Bidzhekov, Tilman M. Hackeng, Christian Weber, Hans-Joachim Gabius, Philipp von Hundelshausen, and Kevin H. Mayo

Review timeline:

Submission date:	1 February 2019
Editorial Decision:	27 February 2019
Revision received:	31 July 2019
Editorial Decision:	20 August 2019
Revision received:	22 October 2019
Editorial Decision:	6 November 2019
Revision received:	20 December 2019
Editorial Decision:	13 January 2020
Revision received:	16 January 2020
Accepted:	20 January 2020

Editor: Achim Breiling

Transaction Report:

1st Editorial Decision

27 February 2019

Thank you for the submission of your research manuscript to EMBO reports. We have now received the reports from the three referees that were asked to evaluate your study, which can be found at the end of this email.

As you will see, all referees think the manuscript is of high interest, but requires major revisions to allow publication here. As the reports are below, and I think all points need to be addressed, I will not further detail them here. Given the constructive referee comments, we would like to invite you to revise your manuscript with the understanding that all referee concerns must be addressed in the revised manuscript and/or in a detailed point-by-point response. Acceptance of your manuscript will depend on a positive outcome of a second round of review.

It is EMBO reports policy to allow a single round of revision only and acceptance or rejection of the manuscript will therefore depend on the completeness of your responses included in the next, final version of the manuscript. Revised manuscripts should be submitted within three months of a request for revision; they will otherwise be treated as new submissions. Please contact me if a 3-months time frame is not sufficient so that we can discuss the revisions further.

Supplementary/additional data: The Expanded View format, which will be displayed in the main HTML of the paper in a collapsible format, has replaced the Supplementary information. You can submit up to 5 images as Expanded View. Please follow the nomenclature Figure EV1, Figure EV2 etc. The figure legend for these should be included in the main manuscript document file in a section

called Expanded View Figure Legends after the main Figure Legends section. Additional Supplementary material should be supplied as a single pdf labeled Appendix. The Appendix includes a table of content on the first page, all figures and their legends. Please follow the nomenclature Appendix Figure Sx throughout the text and also label the figures according to this nomenclature.

For more details please refer to our guide to authors:
<http://embor.embopress.org/authorguide#manuscriptpreparation>

Important: All materials and methods should be included in the main manuscript file.

See also our guide for figure preparation:
http://www.embopress.org/sites/default/files/EMBOPress_Figure_Guidelines_061115.pdf

Regarding data quantification and statistics, can you please specify, where applicable, the number "n" for how many independent experiments (biological replicates) were performed, the bars and error bars (e.g. SEM, SD) and the test used to calculate p-values in the respective figure legends. Please provide statistical testing where applicable. See:
<http://embor.embopress.org/authorguide#statisticalanalysis>

Please also follow our guidelines for the use of living organisms, and the respective reporting guidelines: <http://embor.embopress.org/authorguide#livingorganisms>

We now strongly encourage the publication of original source data with the aim of making primary data more accessible and transparent to the reader. The source data will be published in a separate source data file online along with the accepted manuscript and will be linked to the relevant figure. To use this opportunity, please submit the source data (for example scans of entire gels or blots, data points of graphs in an excel sheet, additional images, etc.) of your key experiments together with the revised manuscript. Please include size markers for scans of entire gels, label the scans with figure and panel number, and send one PDF file per figure.

Please also format the references according to EMBO reports style. See:
<http://embor.embopress.org/authorguide#referencesformat>

- a complete author checklist, which you can download from our author guidelines (<http://embor.embopress.org/authorguide#revision>). Please insert page numbers in the checklist to indicate where the requested information can be found.
- a letter detailing your responses to the referee comments in Word format (.doc)
- a Microsoft Word file (.doc) of the revised manuscript text
- editable TIFF or EPS-formatted single figure files in high resolution (for main figures and EV figures)

Please also note that we now mandate that the corresponding author lists an ORCID digital identifier that is linked to his/her EMBO reports account!

I look forward to seeing a revised version of your manuscript when it is ready. Please let me know if you have questions or comments regarding the revision.

REFEREE REPORTS

Referee #1:

This work by Eckardt and colleagues describes the interaction between galectin-3 and the chemokine CXCL12. They use NMR spectroscopy and computer simulations to deduce a structural model of the heterodimer. The CXCL12:galectin-3 interaction is reduced by two mutations in the galectin-3 polypeptide. The interaction-characteristics are reflected by measuring the CXCL12 chemotaxis in the presence of galectin-3 and galectin-3 mutants. The authors conclude that interaction between chemokines and galectines provides a regulatory mechanism in inflammation.

This is an attractive scenario and would considerably expand our knowledge about galectin functions in inflammation. However, the experiments provided lack essential controls to justify fundamental conclusions made in the paper.

- identification of binding epitopes in Fig. 2 is poorly described. The data should be discussed more comprehensively.

- it is not clear to me why the N-terminally truncated variants of galectin-3 mutants were used in the solid-phase immunoassays and SPR depicted in Appendix Fig. S3. Why does the S-face mutant N160D affect binding to CXCL12? According to the model in Fig. 2E this mutation is quite distant from the interaction motive. The model needs to be checked by a sufficient number of mutations in the F- and S-face of galectin-3 for clarification.

- can the inhibitory effects of galectin-3 on leukocyte migration as depicted in Fig. 3A and B be rescued by increasing concentrations of CXCL12? This needs to be addressed to clarify the involvement of CXCL12 in the galectin-3-dependent migration inhibition.

- DMJ inhibits N-glycosylation, reduces the number of glycans exposed at the cell surface and affects intracellular glycoprotein transport to the cell surface. Thus, DMJ by itself should have inhibitory effects on cell migration. Corresponding control experiments are missing in Appendix Fig. S4.

- The proximity ligation assay depicted in Fig. 4A requires negative and positive controls with matching first antibody combinations.

Referee #2:

In this manuscript, Eckardt and colleagues test the interesting and novel idea that galectins could act as selective chemokine inhibitors focusing mainly on the interaction of CXCL12 with galectin-3. The literature contains many examples of pro-inflammatory actions of galectins, although some examples of anti-inflammatory actions have also been described (e.g. PMID: 21899917) and galectins have been considered as potential anti-inflammatory therapeutics. The current paper adds to this but doesn't settle *in vivo* the physiologic (as opposed to pharmacologic) impact of galectin-3 on CXCL12 function. This would require more than the peritoneal challenge described in Figure 4. Thus, despite the structural data and *in vitro* and challenge functional data, the present paper does not define the biological significance of the chemokine-galectin interactions they describe. In addition, many experiments are lacking essential controls and some others show very small effects, which together with the very limited information provided about how the experiments were performed, leave the reader questioning the main conclusions of this work regarding chemokine-galectin interaction, as detailed below.

MAJOR COMMENTS

1. Some of the interactions the authors show in Figure 1A are very weak. In fact, there are several important disparities when compared to the SPR screening in Figure EV1A that the authors do not discuss. For example, CCL27, CCL24 or CXCL10 are shown as binding partners for Gal-3 in Figure 1A but their interaction by SPR in Figure EV1A is near undetectable. Figure 1A could be better interpreted if the authors included a chemokine strip incubated only with streptavidin-HRP run and developed in parallel to a strip incubated with Gal-3 and Gal-1. Also, although the authors say that SPR experiments confirmed the observations in Figure 1A (page 5, 1st line), in my opinion, this statement is not completely true. The authors should discuss some of the discrepancies mentioned above.

2. The selection of CXCL12 appears to have been quite random. Was there a scientific or technical reason to focus on this chemokine? Is CXCL12 the most likely ligand of Gal-3 *in vivo*? Also, although Gal-3 is able to interact with many chemokines, the authors only study the binding affinity of Gal-3 for CXCL12. It would be very informative to see how this affinity compare to the K_D of Gal-3 for other chemokines.

3. An important negative control is missing in Figure EV1D. In order to confirm that the band the authors highlight in this gel as the Gal-3: CXCL12 complex is the result of a specific binding, the

authors test the same conditions with a protein unable to interact with CXCL12. For example, Gal-3 N160D shown in Figure S3G would make for an excellent negative control. Also, how do the authors explain that BS3 did not crosslink the complex and needed a crosslinker with a much longer spacer arm (BS(PEG)5) to visualize the complex in an SDS-PAGE?

4. At the end of the 2nd paragraph in page 5 the authors say that they could not characterize the interaction of CXCL12 to Gal-3 CRD by SPR because the immobilization could have buried important binding epitopes. However, in the Material and Methods the authors describe that proteins were immobilized using SA chips and biotinylated proteins, a method especially well-suited to avoid this type of interference. One could be concerned that the reason why the authors could not detect this interaction by SPR is simply because the interaction of CXCL12 to Gal-3 CRD is too weak. Since this is a fundamental experiment for the conclusions of the paper, the authors should try to prove this interaction and its affinity by other immobilization methods or other type of assays. In the current form of the manuscript, the experiments that support a direct interaction between Gal-3 CRD and CXCL12 are shown in Fig. EV1 and Fig. S3B. Fig. EV1 is lacking an important negative control as mentioned above. Similarly, Fig. S3B should include a spot of an irrelevant protein unable to interact with CXCL12 or any other negative control. The importance of a proper negative control in this experiment is highlighted by the fact that the binding of CXCL12 to N160D is very similar to N222A whereas in Figure S3G, the binding of the chemokine to N160D is undetectable by SPR.

5. In Fig S3C and S3D the authors extract quantitative conclusions from experiments like the one shown in fig S3C. This type of solid-phase assay is well-suited for acquiring rapid qualitative information, but quantitative conclusions should be approached with caution, especially when essential controls are missing as mentioned above. The experiments in Fig S3G are much more convincing in this regard. Also, in Fig S3C and S3D, the authors do not indicate the number of independent assays/spots used to derive the mean and error values calculated.

6. Fig S3G shows the only SPR sensorgrams included in the manuscript and shows that the dissociation of CXCL12 from Gal-3 is almost instantaneous. Quite often, the stability of an interaction is a much more important parameter than the overall affinity to judge the potential biological significance of an interaction *in vivo*. By the sensorgrams in Fig S3G one could estimate that the $t_{1/2}$ of the Gal-3-CXCL12 complex may be less than a couple of seconds. The authors should discuss this rapid dissociation and be cautious when describing the implications of this interaction *in vivo* in the Discussion.

7. Overall chemotaxis assays are difficult to interpret. The information in the Materials and Methods about how these experiments were performed and analyzed should be expanded. It is not clear how many cells were placed on the top of the filter and how the authors quantified the migrated cells. Also, chemotaxis index (CI) is typically calculated as the ratio between the number of migrated cells and the total number of cells placed on the top of the filter. But here the authors calculated the CI as the ratio of migrated cells between unstimulated and chemokine stimulated cells. Furthermore, the final data are represented as a % CI which in my opinion is an over-processing of the data and hinders the reader's ability to evaluate the robustness of these results. Data in these experiments should be represented as the total number of migrated cells or at the very least as the absolute CI (migrated cells/total cells) for each condition. Also, figure S5D contains essential controls for these experiments and should be moved to the main figure.

8. In Figure S4, the authors nicely show that the inhibitory effect of Gal-3 at 1 μ M is chemokine independent by inhibiting the expression of galectin ligands with 1-DMJ. However, they observe that 1-DMJ also impairs the inhibitory activity of Gal-3 at 0.1 nM. Doesn't this suggest that the effect of Gal-3 at low-doses is also chemokine independent?

9. Figure 4A is missing a control where cells are incubated only with Gal-3 CRD.

10. The BRET ratios shown in Figure 4C are very small. In my opinion, this figure does not completely convince the reader that there are biologically significant differences between 0.035 (response in the absence of Gal-3CRD) and 0.025 (minimum response in the presence of Gal-3). This is aggravated by the fact that an essential control, BRET ratio in the absence of chemokine, is missing. It would be much more convincing if the authors included the same Gal-3 CRD dose

response in the absence of chemokine.

11. Are the in vivo concentrations of galectin-3 compatible with the EC50s for chemokine interaction?

12. SEM is used throughout the manuscript to represent experimental error but most of the graphs represent means of individual data points with $n < 5$. SEM is a valid parameter when graphs contain data from multiple independent experiments, which is not the case in this study. The authors should use SD instead and redo the statistics.

MINOR COMMENTS

1. Page 5, 2nd paragraph, the authors mention that the CXCL12-Gal-3 affinity is "high" but the KD is 100 nM. Chemokine receptors interact with chemokines with a KD about 1-10 nM. Compared to these other proteins the affinity of Gal-3 for chemokines is significantly lower. The authors should tone down this statement.

2. For a better interpretation of the results in Figure 3H and 3I the author should include the gating strategy and an example dot-plot for the staining of monocytes and neutrophils in the peritoneum.

3. To better understand Figures 4F and 4G the authors should include a pertinent example of the acquired images for each condition and describe how the images were processed and analyzed.

----- Referee #3:

The study is novel and uncovers a new mechanism by which galectins modulate leukocyte recruitment. The authors have identified interaction between Gal-3 and CXCL12 that is dependent on the CRD (but not the glycan binding face) of Gal-3. Gal-3 reduces signalling of CXCL12 via its receptor CXCR4 and functionally inhibits CXCL12-dependent chemotaxis. The hypothesis that truncation of full length Gal-3 (which has pro-recruitment properties) into a form that has anti-inflammatory properties is attractive. The manuscript is concise and well written and represents an important advance in the field of galectin biology, with the potential for extension to other members of the galectin family. The manuscript is of interest to researchers within the fields of inflammation and resolution as well as galectin biology.

The experiments performed include appropriate controls such as lactose for assessing carbohydrate-dependent binding as well as assessment of both the full Galectin-3 protein and the CRD domain. The authors have addressed several of my potential queries in the supplementary data for example: aggregation of cells at higher concentrations of galectin.

Comments:

1. The authors should indicate the source of the galectins used. Galectin-1 for example is subject to oxidation and many studies have been published using modified forms of the protein that are resistant to oxidation, can the authors rule out that the lack of a response to Gal-1 compared to Gal-3 is not due to inactivity of the protein?

2. Figure 1A, does the membrane shown show Gal-1 or Gal-3 binding? This should be indicated in the figure legend. It would be informative to show a membrane for each galectin to compare binding.

3. Page 8 The following sentence requires clarification: "Since the affinity of Gal-1 for CXCL12 was low compared to Gal-3, we hypothesised that the inhibition of chemotaxis was caused by physical interaction with the chemokine". Are the authors referring to the effect of Gal-1 at $1 \mu\text{M}$? The data in the supplemental figures suggests that the lack of chemotaxis in response to $1 \mu\text{M}$ Gal-1 was due to aggregation of the cells rather than interaction with the chemokine.

Referee #1:

This work by Eckardt and colleagues describes the interaction between galectin-3 and the chemokine CXCL12. They use NMR spectroscopy and computer simulations to deduce a structural model of the heterodimer. The CXCL12:galectin-3 interaction is reduced by two mutations in the galectin-3 polypeptide. The interaction-characteristics are reflected by measuring the CXCL12 chemotaxis in the presence of galectin-3 and galectin-3 mutants. The authors conclude that interaction between chemokines and galectines provides a regulatory mechanism in inflammation.

This is an attractive scenario and would considerably expand our knowledge about galectin functions in inflammation. However, the experiments provided lack essential controls to justify fundamental conclusions made in the paper.

***Response:** This reviewer listed five points with respect to a lack of essential controls. We addressed these as follows in order “to justify fundamental conclusions”.*

-- identification of binding epitopes in Fig. 2 is poorly described. The data should be discussed more comprehensively.

***Response:** We described the binding epitope within the heterodimer in more detail (please see new paragraph p.8, l. 4-13). For better clarity, we now also show a decomposition analysis per residue for Gal-3 (**Appendix Figure S7A**) and CXCL12 (**Appendix Figure S7B**) and highlight the most important residues in **Appendix Figure S9C**.*

Panel A-D from Figure S7: Decomposition analysis of the CXCL12:Gal-3 CRD heterodimer, model of the heterodimer interface and ΔG of single amino acid mutants of Gal-3 CRD.

-- it is not clear to me why the N-terminally truncated variants of galectin-3 mutants were used in the solid-phase immunoassays and SPR depicted in Appendix Fig. S3. Why does the S-face mutant N160D affect binding to CXCL12? According to the model in Fig. 2E this mutation is quite distant from the interaction motive. The model needs to be checked by a sufficient number of mutations in the F- and S-face of galectin-3 for clarification.

Response: To further validate our structural model of the Gal-3: CXCL12 heterodimer, we experimentally investigated several additional Gal-3 CRD mutants (H217A, Q220E, Q220K,

*N160A, R168A) in complex with CXCL12. S-face mutant Gal-3 CRD^{N160D} (as control) did not result in reconciling the issue as stated by the reviewer; we may explain this as follows: We chose to mutate N160 to demonstrate that a crucial residue within the canonical lactose-binding site (distinct from the CXCL12 contact region) would affect glycan binding to the glycoprotein ASF, but would not affect binding to CXCL12. However, it appears that this mutation also allosterically induced conformational changes in the CRD, as observed with the C2S mutant of Gal-1 and the F19Y mutant of Gal-8 [1, 2]. Indeed, as we showed in the former case (Appendix figure S3G, **now EV3J**) the N160D mutation abolished binding to ASF. However, these data tell us that the affinity for CXCL12 was reduced as well (see Figure S3A). Comparison of MD simulations for interactions of CXCL12 with either Gal-3 CRD (WT) or Gal-3 CRD^{N160D} indicate that reduced affinity might not be a direct consequence of altering residues of the binding epitope, but rather an indirect affect via conformational changes. In this regard, converting asparagine (N) into aspartate (D) introduces a negatively charged residue, which may result in conformational changes and modify H-bonding patterns and electrostatic interactions leading to a more positive binding free energy. E.g. Gal-3 R144 can form an H-bond with D160 in Gal-3 CRD^{N160D} that is not observed in WT Gal-3. Moreover, an H-bond between Gal-3 CRD^{E184} and CXCL12^{N30} is formed via side chain-side chain interactions in Gal-3 CRD^{N160D}, whereas this H-bond is formed by backbone-backbone interactions in the WT lectin, which results in a more stable complex. H-bond interactions between Gal-3 CRD^{E184} and CXCL12^{N30} can influence the 40s loop conformation and also interactions between Gal-3 and the N-terminal region of CXCL12, as observed between Gal-3 CRD^{R183} - CXCL12^{S6} in WT Gal-3 but not in Gal-3 CRD^{N160D}. Our reasoning is based on MD simulations illustrated in the figure below.*

WT

Binding Free Energy (BFE) = -50.39 ± 6.62 kcal/mol

N160D

Binding Free Energy (BFE) = -42.62 ± 8.75 kcal/mol

Figure for the referees: Impact of mutating N-to-D in residue 160 of Gal-3 CRD

All together, these MD simulations highlight the possibility of long-range effects due to a single-site mutation, in this case distant from the site of interaction with CXCL12. Obviously, the presence of an Asp residue causes a drastic change.

In response to the reviewer's concern and to avoid confusion, experiments with Gal-3 CRD^{N160D} have been removed from the revised version of our manuscript. Instead, we produced a neutral Gal-3 CRD^{N160A} mutant that also shows reduced affinity for the cognate sugar, but binds with comparable affinity to CXCL12. Thus, the controversial issue on properties of the N-to-D mutant is removed. Data on this new mutant are shown now in Appendix Figure S8A,B and EV3B,H,J, and described on p.9, l. 6-12.

Appendix Figure S8. Ligand blots of CXCL12 and Gal-3 and Gal-3 CRD mutants.

Expanded View Figure 3

Expanded View Figure 3. Expanded View Figure 3. Binding of Gal-3 mutants to CXCL12 and ASF

We further validated our model by engineering four additional mutants (Q220E, Q220K, H217A, R168A) as explained in the text (p.8, l.25-p.9 l.12), and we examined their binding to CXCL12 obtaining results as illustrated by SPR in **Figure EV3A-L** and NMR in **Figure EV4**.

-- can the inhibitory effects of galectin-3 on leukocyte migration as depicted in Fig. 3A and B be rescued by increasing concentrations of CXCL12? This needs to be addressed to clarify the involvement of CXCL12 in the galectin-3-dependent migration inhibition.

Response: This experiment, suggested by the reviewer, is not as straightforward as it may seem, primarily because the chemotactic effect of CXCL12 follows a bell-shaped curve, and high concentrations of CXCL12 result in reduced cell migration. We therefore compared chemotaxis with increasing concentrations of CXCL12 alone and in combination with Gal-3 CRD (**Figure S10I**). Note that the inhibitory effect of Gal3 CRD increases as the CXCL12 concentration decreases (p. 10, l. 14-20).

Expanded View Figure 4. Formation of a heterodimer between CXCL12 and Gal-3 mutants

Panel I of Appendix Figure S10. Dependence of leukocyte migration on the glycan binding capacity of the galectin.

-- DMJ inhibits N-glycosylation, reduces the number of glycans exposed at the cell surface and affects intracellular glycoprotein transport to the cell surface. Thus, DMJ by itself should have inhibitory effects on cell migration. Corresponding control experiments are missing in Appendix Fig. S4.

Response: We have examined this issue and can now demonstrate (**Appendix Figure S10G**) that DMJ does not significantly inhibit CXCL12-induced Jurkat T cell migration (**p.11, l.18-19**). Moreover, even if we assume that DMJ treatment reduces chemotaxis, this would not

G

Panel G from Appendix Figure S10

alter our conclusions: effects of DMJ treatment on Gal-3 increase the chemotaxis of Jurkat T cells as DMJ attenuates the Gal-3-induced inhibition of CXCL12. Another interesting aspect is that DMJ, by inhibiting α -mannosidase II, switches production from complex-type (Gal-3 binding) to high-mannose-type N-glycans, which can bind to the non-canonical binding site of Gal-3 overlapping with the CXCL12-

binding site [3]. On the other hand, we cannot rule out that Gal-3 has additional functional effects when binding to the cell surface via this non-canonical binding site, which possibly adds another layer of complexity to the system.

-- The proximity ligation assay depicted in Fig. 4A requires negative and positive controls with matching first antibody combinations.

Response: Indeed, these controls are important. Therefore, we have now included the appropriate controls in new **Figure 5E**.

E

Panel E from Figure 5: Inhibition of CXCL12-induced CXCR4 signaling by Gal-3 CRD.

Referee #2:

In this manuscript, Eckardt and colleagues test the interesting and novel idea that galectins could act as selective chemokine inhibitors focusing mainly on the interaction of CXCL12 with galectin-3. The literature contains many examples of pro-inflammatory actions of galectins, although some examples of anti-inflammatory actions have also been described (e.g. PMID: 21899917) [4] and galectins have been considered as potential anti-inflammatory therapeutics. The current paper adds to this but doesn't settle in vivo the physiologic (as opposed to pharmacologic) impact of galectin-3 on CXCL12 function. This would require more than the peritoneal challenge described in Figure 4. Thus, despite the structural data and in vitro and challenge functional data, the present paper does not define the biological significance of the chemokine-galectin interactions they describe. In addition, many experiments are lacking essential controls and some others show very small effects, which together with the very limited information provided about how the experiments were performed, leave the reader questioning the main conclusions of this work regarding chemokine-galectin interaction, as detailed below.

***Response:** This reviewer has listed 12 "MAJOR COMMENTS" and three minor points that have provided us with a clear guideline for work required in our revision. We address each point as follows:*

Expanded View Figure 5

Panel C from Figure 4: Formation of a CXCL12:Gal-3 heterodimer in a murine peritonitis model.

To better address the issue of biological relevance and to distinguish the physiologic role of Gal-3 from its pharmacological impact as requested by the referee, we have now included an additional peritoneal recruitment assays as a model of acute inflammation in Gal-3-knock-out mice and wild type controls. In this model, we found a contribution of the CXCL12 receptor

CXCR4 to the recruitment of classical monocytes into the peritoneum of wild type with thioglycolate (TG), as indicated by pre-injection of the CXCR4 antagonist AMD3465 (Figure 4C). Genetic deficiency of Gal-3 enhanced the intraperitoneal recruitment of classical monocytes in both its CXCR4-dependent and -independent component after injection of TG in Gal-3^{-/-} mice (Figure

4C). We surmise that this increase can be attributed to a lack of formation of an anti-inflammatory heterodimer between CXCL12 and Gal-3 (Figure 4C). In addition we performed antibody-based proximity ligation (PLA, Duolink®) to show that CXCL12 and Gal-3 occur in close proximity on cells recruited to the peritoneum after TG injection, thus allowing for functional interactions under inflammatory conditions (Figure EV5A).

To further substantiate formation of a CXCL12:Gal-3 heterodimer in vivo, we stained Gal-3 and CXCL12 simultaneously in frozen sections of lymph nodes from wild type (Figure EV5B) and CXCL12^{-/-} mice (Figure EV5C). We found a partial co-localization of Gal-3 and

CXCL12 that was mainly detectable at the lymph node capsule, where CXCL12-expressing lymphatic endothelial cells come into close vicinity with sub-

Panels B-D of Figure EV5: Colocalization of CXCL12 and Gal-3 in vivo

capsular sinus macrophages (SSM). Further evidence for a close contact between CXCL12 and Gal-3 *in situ* was obtained by antibody-based PLA of lymph nodes explanted from wild type and CXCL12^{-/-} mice (Figure EV5D). These results indicate that CXCL12 and Gal-3 may come into direct contact under physiological conditions (p.12, l.12- p.13, l.4).

MAJOR COMMENTS

1. Some of the interactions the authors show in Figure 1A are very weak. In fact, there are several important disparities when compared to the SPR screening in Figure EV1A that the authors do not discuss. For example, CCL27, CCL24 or CXCL10 are shown as binding partners for Gal-3 in Figure 1A but their interaction by SPR in Figure EV1A is near undetectable. Figure 1A could be better interpreted if the authors included a chemokine strip incubated only with streptavidin-HRP run and developed in parallel to a strip incubated with Gal-3 and Gal-1. Also, although the authors say that SPR experiments confirmed the observations in Figure 1A (page 5, 1st line), in my opinion, this statement is not completely true. The authors should discuss some of the discrepancies mentioned above.

Response: Yes, we do classify CCL27, CCL24 and CXCL10 in Fig. 1E as binders. However, classifications of a chemokine have been made on a qualitative level. Clearly, it was not our intention to argue that every instance of a chemokine signal will result in formation of heterodimers. Although signals with these chemokines on the membrane are weak, they are apparent and we do see a binding response on the sensor chips, albeit weak. Definitively, an obvious mismatch in the presentation in our original version of our manuscript occurred with CXCL9 which gave a robust binding response by SPR, but no visible signal on the membrane. Of course, we did attempt to determine the affinity of CXCL9 relative to CXCL12. However, we were unable to obtain stable curves and equilibrium binding by SPR, such that these

results remain inconclusive. This has been stated in the text, but will not be shown in the manuscript (p.5 l.5-8). A reason for such disparities is likely that (false-) negative results can occur when a binding epitope is inaccessible due to spatial constraints imposed by the immobilization procedure. Because of this, we used different methods to assess binding interactions. We list an interaction as positive if one method shows a signal.

2. The selection of CXCL12 appears to have been quite random. Was there a scientific or technical reason to focus on this chemokine? Is CXCL12 the most likely ligand of Gal-3 in vivo? Also, although Gal-3 is able to interact with many chemokines, the authors only study the binding affinity of Gal-3 for CXCL12. It would be very informative to see how this affinity compare to the KD of Gal-3 for other chemokines.

***Response:** The points are well taken. Focusing on CXCL12 was based on several scientific and technical considerations (p.5 l.15-22). CXCL12 is known to be a comparatively highly expressed chemokine in many tissues and, as such, could engage in functional interplays with multi-site expressed galectins. To make this point, we draw on an analysis with the expression levels derived from the single cell RNAseq data of 20 organs (lymph nodes were not investigated) deposited as the tabula muris (**Appendix Fig. S1**). Most chemokines that can interact with Gal-3 are barely detectable in any of the tissues and cell types, with the exception of CXCL12. To some extent, CCL5, CXCL14, and CXCL17 are also detectable. CXCL12 is mainly derived from endothelial cells and fibroblasts, whereas Gal-3 is expressed*

mostly by

Figure S1

Appendix Figure S1: Coexpression of Gal-3 and interacting chemokines in distinct organs of the mouse.

macrophages (*Gal-3* also referred to as *Mac(=macrophage)2* antigen). Both immune effectors are co-expressed in the aorta, fat, heart, liver, lung and others. *CCL5*, for comparison, is co-expressed in bone marrow with *Gal-3*, *CXCL14* in the lung and skin, and *CXCL17* only in the lung. Furthermore, *CXCL12* is one of the most extensively studied chemokines that has homeostatic and inflammatory properties that make it an important target in various diseases. Admittedly, technical reasons were as well in favor of selecting this chemokine as we have studied *CXCL12* in the past and established helpful tools.

We agree that it is informative to compare the affinity of CXCL12 for Gal-3 with other chemokines. We describe these data (p.5 I.9-14) and provide experimental curves in **Figure EV2** with kinetic measurements for CCL1, CCL5, CCL22, CCL26, CXCL9, and CXCL11.

Expanded View Figure 2

Expanded View Figure 2. SPR analyses of the heterodimer between chemokines and Gal-3.

3. An important negative control is missing in Figure EV1D. In order to confirm that the band the authors highlight in this gel as the Gal-3: CXCL12 complex is the result of a specific binding, the authors test the same conditions with a protein unable to interact with CXCL12. For example, Gal-3 N160D shown in Figure S3G would make for an excellent negative control. Also, how do the authors explain that BS3 did not crosslink the complex and needed a crosslinker with a much longer spacer arm (BS(PEG)5) to visualize the complex in an SDS-

PAGE?

Response: We have now performed the crosslinking experiment as suggested. For reasons outlined above, we did not work with Gal-3 CRD^{N160D}, but rather we chose Gal-3 CRD^{Q220K}

for this study (**Appendix Figure S9**). In this instance, the heterodimer band appears much weaker than that with the wild type lectin.

The question as to why BS3 did not result in crosslinking, is difficult to answer with certainty. BS(PEG)₅ bridges lysine residues that are 21.7 Å apart, whereas the linker length of BS3 is

11.4 Å. We assume that crosslinking occurred more frequently with BS(PEG)₅ due to its increased flexibility to link Lys pairs than with the compound having a shorter linker.

Residue – CXCL12	Residue – Gal-3	Distance (Å)
K27	K210	13.1
K24	K210	3.7
K43	K210	6.1
K64	K196	15.2
K64	K199	21.2
K68	K196	5.7
K68	K199	14.4

Table: Distances between different Lys (NZ1 atom) – Lys (NZ atom) pairs between CXCL12 and Gal-3

For instance, K1 of CXCL12 and K227 (or K64 and K199) of Gal-3 would be likely candidates for pairing (see table and figure above).

Figure for the referees: Lysine residues of Gal-3: CXCL12

4. At the end of the 2nd paragraph in page 5 the authors say that they could not characterize the interaction of CXCL12 to Gal-3 CRD by SPR because the immobilization could have buried important binding epitopes. However, in the Material and Methods the authors describe that proteins were immobilized using SA chips and biotinylated proteins, a method especially well-suited to avoid this type of interference. One could be concerned that the reason why the authors could not detect this interaction by SPR is simply because the interaction of CXCL12 to Gal-3 CRD is too weak. Since this is a fundamental experiment for the conclusions of the paper, the authors should try to prove this interaction and its affinity by other immobilization methods or other type of assays. In the current form of the manuscript, the experiments that support a direct interaction between Gal-3 CRD and CXCL12 are shown in Fig. EV1 and Fig. S3B. Fig. EV1 is lacking an important negative control as mentioned above. Similarly, Fig. S3B should include a spot of an irrelevant protein unable to interact with CXCL12 or any other negative control. The importance of a proper negative control in this experiment is highlighted by the fact that the binding of CXCL12 to N160D is very similar to N222A whereas in Figure S3G, the binding of the chemokine to N160D is undetectable by SPR.

Response: We are pleased to be able to fully address these concerns of the referee in terms of the interaction between Gal-3 CRD and CXCL12 being too weak to be detected. Therefore, we followed her/his advice to test a new immobilization strategy, a really pleasing experience.

Panel F and G of Figure 1. Physical interaction of Gal-3 and -1 with CC and CXC chemokines.

Notably, Gal-3 contains a single cysteine residue at position 173 ($\beta 5$ strand), which for conjugation on to the C1-sensor chip was modified with thiol groups. The affinities between CXCL12 as analyte, as well as Gal-3 and Gal-3 CRD as ligands, are higher than with the neutravidin-biotin procedure

(Figure 1F and G).

Although we can now document the interaction of Gal-3 CRD and chemokines with SPR, we nonetheless proceeded to perform the crosslinking experiment of CXCL12 with Gal-3 CRD as suggested (, p.9, l.14-16). Please also see response to point 3.

Expanded View Figure 4

Expanded View Figure 4. Formation of a heterodimer between CXCL12 and Gal-3 mutants

To strengthen this point, we obtained HSQC spectra of ^{15}N -labeled CXCL12 in combination with some of our newly generated Gal-3 CRD mutants (as discussed in response to referee 1). Here, we observed smaller chemical shift changes with H217A and Q220K, but not with Q220E in comparison to wild type Gal-3 CRD. These data serve to validate our structural model of the heterodimer (Figure EV4).

Missing controls for the former figure

Appendix S3 are included in Appendix Figure S8 (manuscript p.9, l.6-8).

Most importantly, we need to clarify a misunderstanding: the sensorgrams of the Gal-3 variants depicted in Appendix Figure S3G (now Figure EV3I-L) are not binding curves of galectins to immobilized CXCL12, but rather to the Gal-3-binding glycoprotein ASF. This experiment was a control to evaluate whether mutations of Gal-3 abolished carbohydrate binding. Regrettably, the original caption header reading “Reduced binding of Gal-3 CRD and Gal-3 mutants to CXCL12” was incorrect and has been changed in the revised figure legend.

Panels I-L from Figure EV3: Binding of Gal-3 mutants to CXCL12 and ASF

The observation that mutant Gal-3 CRD^{N160D} has reduced affinity for CXCL12 and the N-to-D exchange is not part of the binding epitope have been explained above. From MD-based free binding energies of various mutants, along with the DC analysis, it seems obvious that the region of interaction is fairly large, with several residues being involved. This is in line with the blot in S3B that shows the remaining extent of binding with the mutants. As a note of caution, work with multi-site mutants increases the risk of modifications to the wild type structure, thus possibly introducing indirect effects and erroneous conclusions.

5. In Fig S3C and S3D the authors extract quantitative conclusions from experiments like the one shown in fig S3C. This type of solid-phase assay is well-suited for acquiring rapid qualitative information, but quantitative conclusions should be approached with caution, especially when essential controls are missing as mentioned above. The experiments in Fig S3G are much more convincing in this regard. Also, in Fig S3C and S3D, the authors do not indicate the number of independent assays/spots used to derive the mean and error values calculated.

Response: We fully agree that data from solid-phase assays involving protein adsorption to a matrix are more qualitative than quantitative. Adding our new SPR and NMR experiments was a means to address this valid concern. The number of independent experiments depicted in former Fig.S3C,D was three.

6. Fig S3G shows the only SPR sensorgrams included in the manuscript and shows that the dissociation of CXCL12 from Gal-3 is almost instantaneous. Quite often, the stability of an interaction is a much more important parameter than the overall affinity to judge the potential biological significance of an interaction in vivo. By the sensorgrams in Fig S3G one could estimate that the t_{1/2} of the Gal-3-CXCL12 complex may be less than a couple of seconds.

The authors should discuss this rapid dissociation and be cautious when describing the implications of this interaction in vivo in the Discussion.

Response: We fully agree that representative examples of sensorgrams are informative for the referees and reader, and therefore, we have provided examples in new **Figure 1, Figure EV2** (see above page 16), **EV3** (see above page 9). We apologize for causing a misunderstanding about Figure S3G. The fast dissociation is related to the interaction with the Gal-3 binding glycoprotein ASF (not CXCL12) and is in line with the literature [5].

Panels F-H from Figure 1: Physical interaction of Gal-3 and -1 with CC and CXC chemokines.

7. Overall chemotaxis assays are difficult to interpret. The information in the Materials and Methods about how these experiments were performed and analyzed should be expanded. It is not clear how many cells were placed on the top of the filter and how the authors quantified the migrated cells. Also, chemotaxis index (CI) is typically calculated as the ratio between the number of migrated cells and the total number of cells placed on the top of the filter. But here the authors calculated the CI as the ratio of migrated cells between unstimulated and chemokine stimulated cells. Furthermore, the final data are represented as a % CI which in my opinion is an over-processing of the data and hinders the reader's ability to evaluate the robustness of these results. Data in these experiments should be represented as the total number of migrated cells or at the very least as the absolute CI (migrated cells/total cells) for each condition. Also, figure S5D contains essential controls for these experiments and should be moved to the main figure.

Response: We have described the chemotaxis experiments in greater detail in Material and Methods (p.19, 1.19-26). In every chemotaxis experiment, we used the same number of cells (10^5) that were placed on top of the filter. Since the migratory behavior of primary cells, as

well as of Jurkat T cells, differed towards CXCL12, we normalized the number of migrating cells to CXCL12. This value was set to 100% in order not to overly weight an individual experiment. We have also indicated the absolute values as requested by the referee and recalculated all the statistics. Values are now indicated as mean \pm SD throughout the manuscript.

8. In Figure S4, the authors nicely show that the inhibitory effect of Gal-3 at 1 μ M is chemokine independent by inhibiting the expression of galectin ligands with 1-DMJ. However, they observe that 1-DMJ also impairs the inhibitory activity of Gal-3 at 0.1 nM. Doesn't this suggest that the effect of Gal-3 at low-doses is also chemokine independent?

Response: To further rule out that at lower concentrations, CXCL12-independent Gal-3

H

Appendix Figure S10 panel H: Dependence of leukocyte migration on the glycan binding capacity of the galectin.

effects play a major role, we tested a small molecule CXCR4 agonist that binds allosterically (NUCC-390). This experiment allowed us to confirm that this molecule can induce chemotaxis [6]. Neither Gal-3 nor Gal-3 CRD inhibited NUCC-390-mediated chemotaxis so that under these conditions chemokine-independent effects definitely do not occur (**Appendix Figure S10H, p.14, l.10-14**). However, we stated in the manuscript that chemokine-dependent and -independent effects may well occur simultaneously for full-length, wild type Gal-3 (**p. 11, l. 8-12**).

9. Figure 4A is missing a control where cells are incubated only with Gal-3 CRD.

Response: We included the requested control, now **Figure 5A** (the Gi-signaling experiment). Of note, the Gal-3 CRD alone does not affect cAMP levels.

A

Panel A from Figure 5:

Inhibition of CXCL12-induced CXCR4 signaling by Gal-3 CRD.

10. The BRET ratios shown in Figure 4C are very small. In my opinion, this figure does not completely convince the reader that there are biologically significant differences between 0.035 (response in the absence of Gal-3CRD) and 0.025 (minimum response in the presence of Gal-3). This is aggravated by the fact that an essential control, BRET ratio in the absence of chemokine, is missing. It would be much more convincing if the authors included the same Gal-3 CRD dose response in the absence of chemokine.

Response: We measured the effect of increasing concentrations of Gal-3 CRD on β -arrestin recruitment (BRET ratio) in the absence of CXCL12 (Fig. 5B,C). Gal-3 CRD alone did not trigger the effect at any concentration β -arrestin-2 recruitment to CXCR4.

Panels B and C from Figure 5: Inhibition of CXCL12-induced CXCR4 signaling by Gal-3 CRD.

11. Are the in vivo concentrations of galectin-3 compatible with the EC50s for chemokine interaction?

Response: This question is difficult to answer with 100% confidence, because one can only compare EC50 values that are measured in the same system. The EC50 values of the chemokine interaction are reflected by K_D values measured by SPR which are roughly 100 nM for Gal-3: CXCL12 and a stoichiometry of 1:100 to 1:10 required to inhibit chemotaxis by 50%. After the i.p. injection, it is impossible to assess the profile and dynamics of exchange of concentrations with increasing cell and fluid influx that both occur in such a complex environment. We injected a solution of 500 nM CXCL12 to trigger leukocyte recruitment and were able to block this by using a combination with 50 nM Gal-3. Therefore, we believe that it

is reasonable to assume that the Gal-3 concentration used in this in vivo experiment can be reconciled with the effects seen in our in vitro experiments.

12. SEM is used throughout the manuscript to represent experimental error but most of the graphs represent means of individual data points with $n < 5$. SEM is a valid parameter when graphs contain data from multiple independent experiments, which is not the case in this study. The authors should use SD instead and redo the statistics.

Response: Values are now indicated as mean \pm SD throughout the manuscript, and all statistics have been recalculated, as also addressed in point 7 above.

MINOR COMMENTS

1. Page 5, 2nd paragraph, the authors mention that the CXCL12-Gal-3 affinity is "high" but the KD is 100 nM. Chemokine receptors interact with chemokines with a KD about 1-10 nM. Compared to these other proteins the affinity of Gal-3 for chemokines is significantly lower. The authors should tone down this statement.

Response: We modified this by using the word "remarkable" on p.5, l.24.

2. For a better interpretation of the results in Figure 3H and 3I the author should include the gating strategy and an example dot-plot for the staining of monocytes and neutrophils in the peritoneum.

Response: The gating strategy is explained in the new **Appendix Fig. S12**.

Appendix Figure S12

Appendix Figure S12. Gating strategy of leukocyte staining in peritoneal lavage.

3. To better understand Figures 4F and 4G the authors should include a pertinent example of the acquired images for each condition and describe how the images were processed and analyzed.

Response: The data were acquired by in measurements using a flow cytometer and not by a microscope.

Referee #3:

The study is novel and uncovers a new mechanism by which galectins modulate leukocyte recruitment. The authors have identified interaction between Gal-3 and CXCL12 that is dependent on the CRD (but not the glycan binding face) of Gal-3. Gal-3 reduces signalling of CXCL12 via its receptor CXCR4 and functionally inhibits CXCL12-dependent chemotaxis. The hypothesis that truncation of full length Gal-3 (which has pro-recruitment properties) into a form that has anti-inflammatory properties is attractive. The manuscript is concise and well written and represents an important advance in the field of galectin biology, with the potential for extension to other members of the galectin family. The manuscript is of interest to researchers within the fields of inflammation and resolution as well as galectin biology.

The experiments performed include appropriate controls such as lactose for assessing carbohydrate-dependent binding as well as assessment of both the full Galectin-3 protein and the CRD domain. The authors have addressed several of my potential queries in the supplementary data for example: aggregation of cells at higher concentrations of galectin.

COMMENTS:

1. The authors should indicate the source of the galectins used. Galectin-1 for example is subject to oxidation and many studies have been published using modified forms of the protein that are resistant to oxidation, can the authors rule out that the lack of a response to Gal-1 compared to Gal-3 is not due to inactivity of the protein?

Response: Indeed, this point requires explanation. Galectin-1 is special among galectins due to its relatively rapid inactivation in an oxidative environment. Concerns have been repeatedly raised in the literature that this galectin be studied in the presence of a reducing

agent, i.e. β -mercaptoethanol or dithiothreitol. However, these agents can have an effect on cells. It is an accepted and common procedure to protect Gal-1 surface Cys residues, especially Cys2, Cys16, and Cys130, prone to oxidative damage by carboxamido methylation using iodoacetamide as introduced by P.L. Whitney et al. [7]. Galectin-1 then remains fully active in functional assays, alone and in mixtures, showing functional antagonism and cooperation [8-10]. The procedure and evidence for functional integrity are now given by these references. We should add that involvement of Cys residues in intermolecular interactions between Gal-3 and CXCL12 is not an issue here, considering the contact region described for Gal-3 and CXCL12 above.

2. Figure 1A, does the membrane shown show Gal-1 or Gal-3 binding? This should be indicated in the figure legend. It would be informative to show a membrane for each galectin to compare binding.

Response: In Figure 1, the representative membrane was incubated with Gal-3-containing solution; another membrane treated with Gal-1 is shown in **Figure EV1**.

Expanded View Figure 1

Panels A and B from Figure EV1. Physical interaction of Gal-3 and -1 with CC and CXC chemokines.

3. Page 8 The following sentence requires clarification: "Since the affinity of Gal-1 for CXCL12 was low compared to Gal-3, we hypothesized that the inhibition of chemotaxis was caused by physical interaction with the chemokine". Are the authors referring to the effect of Gal-1 at 1 μ M? The data in the supplemental figures suggests that the lack of chemotaxis in response to 1 μ M Gal-1 was due to aggregation of the cells rather than interaction with the chemokine.

Response: To provide clarification on this point, we revised the text (p.10, l.14-20). Here, we correlated low affinity and low functional effects of Gal-1 at sub-micromolar concentrations when aggregation does not occur.

References

1. Lopez-Lucendo MF, Solis D, Andre S, Hirabayashi J, Kasai K, Kaltner H, Gabius HJ, Romero A (2004) Growth-regulatory human galectin-1: crystallographic characterisation of the structural changes induced by single-site mutations and their impact on the thermodynamics of ligand binding. *Journal of molecular biology* **343**: 957-70
2. Ruiz FM, Scholz BA, Buzamet E, Kopitz J, André S, Menéndez M, Romero A, Solís D, Gabius H-J (2014) Natural single amino acid polymorphism (F19Y) in human galectin-8: detection of structural alterations and increased growth-regulatory activity on tumor cells. *The FEBS Journal* **281**: 1446-1464
3. Miller MC, Ippel H, Suylen D, Klyosov AA, Traber PG, Hackeng T, Mayo KH (2016) Binding of polysaccharides to human galectin-3 at a noncanonical site in its carbohydrate recognition domain. *Glycobiology* **26**: 88-99
4. Baseras B, Gaida MM, Kahle N, Schuppel AK, Kathrey D, Prior B, Wente M, Hansch GM (2012) Galectin-3 inhibits the chemotaxis of human polymorphonuclear neutrophils in vitro. *Immunobiology* **217**: 83-90
5. Bocker S, Elling L (2017) Binding characteristics of galectin-3 fusion proteins. *Glycobiology* **27**: 457-468
6. Mishra RK, Shum AK, Plataniias LC, Miller RJ, Schiltz GE (2016) Discovery and characterization of novel small-molecule CXCR4 receptor agonists and antagonists. *Sci Rep* **6**: 30155
7. Whitney PL, Powell JT, Sanford GL (1986) Oxidation and chemical modification of lung beta-galactoside-specific lectin. *Biochem J* **238**: 683-9
8. Amano M, Eriksson H, Manning JC, Detjen KM, Andre S, Nishimura S, Lehtio J, Gabius HJ (2012) Tumour suppressor p16(INK4a) - anoikis-favouring decrease in N/O-glycan/cell surface sialylation by down-regulation of enzymes in sialic acid biosynthesis in tandem in a pancreatic carcinoma model. *FEBS J* **279**: 4062-80
9. Toegel S, Weinmann D, Andre S, Walzer SM, Bilban M, Schmidt S, Chiari C, Windhager R, Krall C, Bennani-Baiti IM, *et al.* (2016) Galectin-1 couples glycobiology to inflammation in osteoarthritis through the activation of an NF- κ B-regulated gene network. *J Immunol* **196**: 1910-21
10. Weinmann D, Schlangen K, Andre S, Schmidt S, Walzer SM, Kubista B, Windhager R, Toegel S, Gabius HJ (2016) Galectin-3 induces a pro-degradative/inflammatory gene signature in human chondrocytes, teaming up with galectin-1 in osteoarthritis pathogenesis. *Sci Rep* **6**: 39112

2nd Editorial Decision

20 August 2019

Thank you for the submission of your revised manuscript to our editorial offices. We have now received the reports from the three referees that were asked to re-evaluate your study, you will find below. As you will see, the referees now support the publication of your manuscript in EMBO reports. However, referee #2 has a couple of remaining concerns and further suggestions we ask you to address in a final revised version of your manuscript.

- We allow not more than 5 EV figures. Please re-arrange the EV figures, maybe adding some panels to the main figures or to Appendix figures, to have not more than 5 EV figures in the final version of the manuscript.
- Please add page numbers to the TOC of the Appendix, and name the final file 'Appendix'.
- Please add separate call-outs for the panels A-D of Fig. EV 1.
- Fig. EV 2A,B,D,F are presently called out after Fig. EV6. Please call out the panels/figures in a sequential manner. Thus, please re-arrange the figures/panels and/or the manuscript text.
- Please make sure that where applicable, the number "n" for how many independent experiments (biological or technical replicates - please specify) were performed, the bars and error bars (e.g. SEM, SD) and the test used to calculate p-values in the respective figure legends are indicated. Please provide statistical testing where applicable. See also point 1 of referee #2.
- Please remove any writing from the scale bars. Please indicate the size only in the respective figure legend.
- Please move the references before the figure legends in the manuscript text file.
- Please find attached a word file of the manuscript text (provided by our publisher) with changes we ask you to include in your final manuscript text, and some queries, we ask you to address. Please provide your final manuscript file with track changes, in order that we can see the modifications done.

In addition I would need from you:

- a short, two-sentence summary of the manuscript
- two to three bullet points highlighting the key findings of your study
- a schematic summary figure (in jpeg or tiff format with the exact width of 550 pixels and a height of not more than 400 pixels) that can be used as a visual synopsis on our website.

REFeree REPORTS

Referee #1:

My issues were adequately addressed in the revised manuscript.

Referee #2:

This revised manuscript still fails to convincingly demonstrated a functional role for galectins in

regulating chemokine function. The structural aspects of the work appear to be sound. Specific problems are as follows:

1. The reproducibility of the functional data is not clear. The figures should indicate whether the results shown are representative experiments or a summary of multiple independent experiments. The number of experiments performed and the number of replicates in each experiment should be indicated.
2. The analysis of chemotaxis inhibition involves four different cell types, but the great majority of data shown is from a cultured cell line, Jurkat. The experiments with primary leukocytes are limited in scope, the reproducibility is uncertain and the inhibitory effects appear weak. In particular, Figure 3, which is devoted to chemotaxis, only has one panel, panel D, devoted to primary cells, CD4+ T cells, and only for CCL17, not for CXCL12, the major chemokine of interest in the manuscript. Instead CD4+ T cell and neutrophil data are shown in Supplemental Figure 11, and there the scope is limited to one concentration of chemokine, one concentration of galectin. Also, the effects of galectin-1 which is used as a negative control since it binds to CXCL12 very weakly compared to galectin-3 do not appear to be very different from galectin-3. Overall, the inhibitory effects of galectin-3 are weak in this figure and not convincing. Also, how is it that galectin-1 at one micromolar does not inhibit cell migration for these cell types when it completely inhibits migration of CXCL12-stimulated Jurkat cells, probably due to the aggregation effect the authors document?
3. In Figure 3, the controls for panel A are distributed among separate panels. It isn't clear whether these results are all from the same experiment or summaries of all experiments or just representative experiments from different experiments. Given that the chemotactic activity in this assay is very weak (only about 3-5% of input cells migrate to CXCL12), and the variance is large, this representation of the results is not at all convincing. Moreover, in panel 3A there appears to be maximal inhibition by galectin 3 occurring at concentrations that are 100 fold below the Kd. How do the authors explain this? The 100% inhibition at 1 micromolar probably is occurring by some other mechanism than simple binding of galectin to CXCL12, maybe aggregation of the cells as the authors suggest and provide some evidence for in the Appendix.
4. In Figure 3B, the dose response provides weak evidence for a significant functional interaction of the CRD with CXCL12. Specifically, there is an ~50% inhibition of CXCL12-induced chemotaxis from baseline conferred by 1 pM galectin-3 CRD, a concentration that is 30,000-fold lower than the Kd of CXCL12-CRD interaction. Then, there is only ~25% additional inhibition of chemotaxis that takes place incrementally over the next 6 logs of galectin concentration. The conclusion that this inhibitory activity is the consequence of direct CXCL12-CRD binding is not convincing.
5. Figure 4:
 - a. how is it possible for a 10:1 molar excess of CXCL12 over galectin-3 to result in reduced classical monocyte recruitment to peritoneum compared to uninhibited CXCL12? The binding data indicate that the binding affinity of gal-3 to CXCL12 is relatively weak, not remarkable as the authors describe it, ~58 nM Kd. Even if 100% of the gal-3 present were bound to CXCL12, at least 90% of the CXCL12 would remain unbound and presumably active in this experiment.
 - b. In panel C, classical monocyte recruitment by thioglycollate was greater than classical monocyte recruitment by CXCL12 and as great as neutrophil recruitment by CXCL12, a pattern that is unlikely given the cells were harvested from the peritoneal cavity after only a few hours after instillation of the stimulus. At this time point, TG recruits mainly neutrophils, in great excess over monocytes.
 - c. Also in Panel C, it is very surprising that over only 4 hours 50% of classical monocyte recruitment in response to TG is CXCR4-dependent. Much more likely are pro-inflammatory CXC or CC chemokines that are not inhibited by AMD. To make this convincing the authors would need to show that TG induces CXCL12 concentration in the peritoneal cavity. They would also need to reproduce the experiment, which appears to have been performed only once, with 3-4 mice per condition.
 - d. Assuming they are right, that there is a substantial CXCR4-dependent component to TG-induced classical monocyte recruitment, the restoration of this activity in AMD pretreated mice by genetic ablation of gal-3 can have multiple interpretations, including enhance CC or CXC chemokine production in the model. In this regard, why would eliminating a putative inhibitor of CXCL12 in

the Gal-3 knockouts result in restoration of CXCR4-dependent activity after TG when CXCR4 has been fully blocked by the pharmacologic inhibitor of CXCR4 AMD? This is not consistent with the proposed role of Gal-3.

e. In the related Figure EV5A, the authors claim that TG induced recruitment of cells to the peritoneum that are PLA proximity assay positive. However, at most 10% of the cells are positive, and the cells are not identified. Do the PLA+ cells represent 50% of the classical monocytes? If not, how do the authors explain the results in panel B?

6. Figure S10I: there is no negative control. This is important in general, but moreso in particular since the chemotactic index is very low in this system. Does 100 nM gal-1 also inhibit across this concentration range, for example?

Referee #3:

The authors have performed several further experiments to address the concerns raised by the reviewers. This has significantly strengthened the manuscript.

2nd Revision - authors' response

22 October 2019

Point-by-point responses to referee #2:

Point 1 -- Statement by Reviewer 2: "The reproducibility of the functional data is not clear. The figures should indicate whether the results shown are representative experiments or a summary of multiple independent experiments. The number of experiments performed and the number of replicates in each experiment should be indicated."

Response: *This is, of course, an essential point that truly deserves utmost attention (we sincerely regret to have so far been less than optimal in our documentation). Basically, the number of experiments "n" with the functional data (chemotaxis and peritoneal recruitment) is for several independent experiments. In this regard, chemotaxis data are reported on multiple independent experiments, each performed in triplicate; thus "n" indicates the number of independent experiments performed on different days. This is now stated in the Methods Section (p.19, 1.23-24). For the peritoneal recruitment assay, each data point represents values from a single mouse, with "n" standing for the number of mice used in that particular experiment.*

Point 2 -- Statement by Reviewer 2: "The analysis of chemotaxis inhibition involves four different cell types, but the great majority of data shown is from a cultured cell line, Jurkat. The experiments with primary leukocytes are limited in scope, the reproducibility is uncertain and the inhibitory effects appear weak. In particular, Figure 3, which is devoted to chemotaxis, only has one panel, panel D, devoted to primary cells, CD4+ T cells, and only for CCL17, not for CXCL12, the major chemokine of interest in the manuscript. Instead CD4+ T cell and neutrophil data are shown in Supplemental Figure 11, and there the scope is limited to one concentration of chemokine, one concentration of galectin. Also, the effects of galectin-1 which is used as a negative control since it binds to CXCL12 very weakly compared to galectin-3 do not appear to be very different from galectin-3. Overall, the inhibitory effects of galectin-3 are weak in this figure and not convincing. Also, how is it that galectin-1 at one micromolar does not inhibitor cell migration for these cell types when it completely inhibits migration of CXCL12-stimulated Jurkat cells, probably due to the aggregation effect the authors document?"

Response: We performed the majority of our chemotaxis experiments such as those on dose escalation with galectins using Jurkat T cells, because they are as a standard model accessible for all colleagues, thus allowing work in different labs, can be produced in relatively large numbers, and give a robust readout in this assay. With these initial findings in hand, we then proceeded to test validation of those findings using primary cell types, e.g. primary T cells. Native primary T cells are only migrating towards chemokines to a sufficient extent in our hands, when they are cultured and activated for over almost a week, and each primary T cell prep provides only a limited number of cells. With Jurkat cells, we could more readily and easily perform numerous cell-based experiments over a broad range of chemokine/galectin concentrations. Given that the experiments with primary cells appear to generally support our observations with Jurkats, our approach appears to have some merit. However, to stress the importance of primary cell readouts, we did move panel B showing the chemotaxis of primary human T cells towards CXCL12 from former Appendix Figure S11 to a more prominent place in the manuscript, namely Figure 3 (now Fig. 3D).
New Figure 3: Inhibition of CXCL12-induced leukocyte migration by Gal-3 in vitro.

To address the issue whether Gal-1-induced effects differ from those of Gal-3, we have included statistical comparisons between data sets for Gal-1 and for Gal-3 (new Fig. 3D and new Appendix Fig. S11 E, O, P) that show, in most instances, an inhibitory potency of Gal-3 and Gal-3 CRD to be

Figure 3

significantly higher than that of Gal-1. Human neutrophils provide an apparent exception (Appendix Fig. S11P); with this primary cell type, both Gal-3 and the Gal-3 CRD had a relatively weak inhibitory effect (albeit significant), whereas Gal-1 had a slightly inhibitory effect. In this instance, there was no statistical difference with these galectins. We assume that differences in cell type and donor may play a role here. Moreover, we understand that one can view Gal-1 as a negative control, because it does not appear to antagonize CXCL12 over most concentrations tested (see Fig. 3C). The observation that (in several experiments) the presence of Gal-1 does lead to an effect on CXCL12-mediated cell migration is consistent with the interaction between CXCL12 and Gal-1 being weak. In our context, CCL17 (that we identified as a non-binder for galectin-1 and -3) appears to constitute a negative control, because neither Gal-1 nor Gal-3 have any effect on CCL17-induced chemotaxis (Fig. 3E).

Panels E, O and P from new Appendix Figure S11: Effect of aggregation, glycans, and cell viability on leukocyte migration.

We concur with referee 2 that at $1 \mu\text{M}$ and higher concentrations, Gal-1-dependent cell aggregation likely explains its lack of activity to promote cell migration. However, only in dose-escalation chemotaxis experiments (Figures 3 A-C and S11I) did we use a high concentration ($1 \mu\text{M}$) of galectins. The galectin concentration used in the experiments of concern (new Figure 3D, new Appendix Figure S11 E, O, P) was performed at 10 nM (see figure legend p.25, 1.5, Appendix p.19, 1.5-6 and 19). To emphasize this issue, we now also show a data point at 0.1 nM Gal-1, and also the control with lactose to show that this galectin has no effect on CXCL12-mediated cell migration at low concentration (Appendix Figure S11B, Appendix p.18, 1.6, manuscript p. 11, 1. 11 and 16).

Panels A-L from new Appendix Figure S11: Effects of Gal-3 and -1 on aggregation and chemotaxis towards CXCL12.

Point 3 -- Statement by Reviewer 2: "In Figure 3, the controls for panel A are distributed among separate panels. It isn't clear whether these results are all from the same experiment or summaries of all experiments or just representative experiments from different experiments. Given that that the chemotactic activity in this assay is very weak (only about 3-5% of input cells migrate to CXCL12), and the variance is large, this representation of the results is not at all convincing. Moreover, in panel 3A there appears to be maximal inhibition by galectin 3 occurring at concentrations that are 100 fold below the K_d . How do the authors explain this? The 100% inhibition at $1 \mu\text{M}$ probably is occurring by some other mechanism than simple binding of

galectin to CXCL12, maybe aggregation of the cells as the authors suggest and provide some evidence for in the Appendix.”

Response: Controls (PBS, CXCL12 alone) were routinely carried out along with each independent experiment. Galectins alone were tested separately (shown in Fig. 3J), with CXCL12 as a positive control (data not shown).

Regarding the comment on low chemotactic activity, we apologize to have not provided the adequate explanation to preclude the given interpretation of a relationship of percentage to cell number. We used a high input number of cells (100,000 cells/filter, 96-well format) to ensure that the number of cells was not rate limiting in terms of the number of cells migrating to the bottom chamber. For example, if we had used 200,000 cells, the same number of cells would have migrated through the filter; however, the percentage of migrating cells would have been calculated (# cells in bottom chamber divided by # cells in the top chamber times 100) to be even less: that is why we did not report this parameter as such. Under these conditions, the chemotactic index (# of migrating cells) at 10 nM CXCL12 was more than 30 times that of CXCL12 in the presence of galectins: observation of dose-response curves support, the way in which we reported cell migration.

We agree with the referee that the functionally effective concentrations of Gal-3 are much lower than the K_d as measured by SPR and report the data as such. Experience with multivalent ligand presentation and galectins has shown us before that the initial steps of binding can have about a 6000-fold higher affinity than when moving towards saturation (Dam et al., *Biochemistry* 2005, 44:12564-71). This phenomenon of fractional affinities along steps of increasing binding-site saturation may work on the cell surface – but it is too early to postulate this (in principle reasonable) suggestion. Cell surface binding (at high affinity) to a glycoconjugate (with impact on CXCL12 activity as (at least contributing) factor) can not be excluded on the basis of our data. Thus, we have modified the respective text passage in the spirit of this comment, just to be on the safe side (p.16, 1.2-11).

We also agree with referee #2 that 100% inhibition at 1 micromolar Gal-1 probably occurs via some alternative mechanism, likely aggregation. In addition to the experiments that quantify an aggregation (Appendix Figure S11A), we have now provided microscopic images of Jurkat T cells incubated with and without Gal-1, Gal-3 and Gal-3 CRD at a concentration of 1 μ M in order to visualize aggregate formation induced by Gal-1 and by full-length Gal-3, but not, as expected, by Gal-3 CRD.

Figure for referee 2: Light microscopy of Jurkat T cells incubated without (upper left panel), Gal-1 at 1 μ M (upper right panel), Gal-3 at 1 μ M (lower left panel) and Gal-3 CRD at 1 μ M (lower right panel)

Point 4 -- Statement by Reviewer 2: “In Figure 3B, the dose response provides weak evidence for a significant functional interaction of the CRD with CXCL12. Specifically, there is an ~50% inhibition of CXCL12-induced chemotaxis from baseline conferred by 1 pM galectin-3 CRD, a concentration that is 30,000-fold lower than the K_d of CXCL12-CRD interaction. Then, there is only ~25% additional inhibition of chemotaxis that takes place incrementally over the next 6 logs of

galectin concentration. The conclusion that this inhibitory activity is the consequence of direct CXCL12-CRD binding is not convincing.”

Response: First, noting that chemokines and galectins have been considered as separate classes of chemoattractants (for Gal-3 discovered in a pioneering study by Sano et al., *J. Immunol.* 2000;165:2156-64), it is *sui generis* a noteworthy finding to see inhibition. We agree to tone down the direct relationship to the pairing shown in the structural part of our report, as also noted in the previous paragraph.

Point 5a-e -- Statements by Reviewer 2 (concerning Figure 4):

a. “How is it possible for a 10:1 molar excess of CXCL12 over galectin-3 to result in reduced classical monocyte recruitment to peritoneum compared to uninhibited CXCL12? The binding data indicate that the binding affinity of gal-3 to CXCL12 is relatively weak, not remarkable as the authors describe it, ~58 nM K_d. Even if 100% of the gal-3 present were bound to CXCL12, at least 90% of the CXCL12 would remain unbound and presumably active in this experiment.”

Response: This comment follows to the reasoning as given for points 3 and 4 above. Since the reviewer explicitly points to “at least 90% of the CXCL12” being unbound, it is noteworthy that the effective bioactivity of the chemokine is assumed to be affected by interactions with glycosaminoglycans so that only a fraction thereof is bioavailable for the receptor binding (please see Graham et al., *Trends Immunol.* 2019;40:472-81 for details). In fact, to reveal a masking of CXCL12 by heparin, we note that the addition of Gal-3 CRD to heparin-bound CXCL12 did not result in formation of Gal-3 CRD: CXCL12 heterodimers, as illustrated by our results below with SPR chip-conjugated heparin and interaction analysis with CXCL12 and/or Gal-3 CRD (please see figure below).

Figure for referee 2: Solution containing Gal-3 CRD was injected to flow over a chip presenting heparin either alone or in complex with CXCL12 (100 nM). The blue line indicates the value for 100 nM CXCL12 alone.

Following this reasoning, we propose the possibility that while a majority of CXCL12 is bound to glycosaminoglycans on the cell surface and is therefore likely inaccessible to Gal-3, a small number of CXCL12 remains free for receptor activation and is accessible for interactions with Gal-3. In support of this, CXCL12 residues Lys24, His25 and Lys27 are crucial for interactions both with GAGs (Proudfoot et al., *Pharmaceuticals (Basel)* 2017; 10, 70 and Ziarek et al., *J Biol Chem* 2013; 288: 737-746) and with Gal-3 (e.g. Fig. 2A,C). In addition to the figure above, we show new SPR data that demonstrate that incubating heparin with CXCL12 is a means to block binding to Gal-3 CRD (Figure Appendix S2C).

New Appendix Figure S2C: Increasing concentrations of CXCL12 alone (black traces) or mixed with 0.5 $\mu\text{g/ml}$ unfractionated heparin (purple traces) were injected over a Gal-3 CRD presenting sensor chip (three independent experiments).

As an analogy, it has been reported that the inhibitory activity of monoclonal antibodies (m Abs) against CXCL10 depends on whether these mAbs recognize only the free form of the chemokine or also the chemokine in its GAG-bound form Bonvin et al. *J Biol Chem* 2017; 292, 4185-97. In our view, this scenario could explain why small amounts of Gal-3 are sufficient to block CXCL12 at higher concentrations. However, we accept the fact that other explanations may also be possible. For instance, the half-life of Gal-3 and CXCL12 in the peritoneal cavity may differ, because distribution, uptake, and degradation may vary, and the concentration of CXCL12 could fall faster than that of Gal-3.

b. “In panel C, classical monocyte recruitment by thioglycolate was greater than classical monocyte recruitment by CXCL12 and as great as neutrophil recruitment by CXCL12, a pattern that is unlikely given the cells were harvested from the peritoneal cavity after only a few hours after instillation of the stimulus. At this time point, TG recruits mainly neutrophils, in great excess over monocytes.”

Response: We agree with the referee that neutrophils are recruited at early time points (few hours) after TG stimulation. However, at later time points, monocytes, macrophages, and lymphocytes become the dominant populations (see e.g. the figure taken from Hermida et al., *BMC Res Notes* 2017; 10: 695-695). Having used two different time points for CXCL12- and TG-mediated leukocyte recruitment (pilot experiments provided evidence for a later time point than 4 hours): data from Fig.4A,B show recruitment 4 hours post-CXCL12 injection, whereas in panel C, monocyte recruitment was measured 18 hours post-injection of TG. In our previous version, we only described this in the Methods Section, which may have led to misunderstanding; we apologize for this. In the present version, we provided additional information for the reader in the figure legend (p.25, l. 25).

Figure for referee 2 taken from Hermida et al. BMC Res Notes (2017): Kinetics of leukocyte recruitment into the peritoneum after TG stimulation

c. “Also in Panel C, it is very surprising that over only 4 hours 50% of classical monocyte recruitment in response to TG is CXCR4-dependent. Much more likely are pro-inflammatory CXC or CC chemokines that are not inhibited by AMD. To make this convincing the authors would need to show that TG induces CXCL12 concentration in the peritoneal cavity. They would also need to reproduce the experiment, which appears to have been performed only once, with 3-4 mice per condition.”

Response: In panel C, the TG response was measured after a longer time period (18h) and not 4h, during which multiple inflammatory mediators are presumably involved.

To address concerns of reproducibility, we repeated and reproduced the animal experiments in Figure 4 A-C, thus increasing the number of mice per condition and lowering SD values to some extent. In total, statistically significant differences remain as previously reported.

Figure 4

Figure 4: Formation of CXCL12/Gal-3 heterodimer in a murine peritonitis model.

Expanded View Figure 4

New Expanded View Figure 4: Redistribution of CXCL12 from the peritoneal tissue to the lavage and CXCR4 expression levels on monocytes in blood and the peritoneum 18h after TG stimulation. We agree with referee 2 that aside from CXCL12, other chemokines could also play a role. However, the effect of the CXCR4 antagonist provides evidence that the CXCL12-CXCR4 axis will at least contribute to this process. With these new data, the difference between control and AMD has been increased significantly. As requested, we measured CXCL12 levels in the peritoneal tissue and the lavage, and we found a significant shift into the lavage (new Figure EV4A) 18 hours after TG injection. This is consistent with our new data demonstrating that CXCR4 expression on classical blood monocytes is higher compared to classical monocytes in the peritoneal lavage. This is in line with monocytes arising from the blood by CXCL12-induced CXCR4 activation (Fig. EV4B).

d. “Assuming they are right, that there is a substantial CXCR4-dependent component to TG-induced classical monocyte recruitment; the restoration of this activity in AMD pretreated mice by genetic ablation of gal-3 can have multiple interpretations, including enhance CC or CXC chemokine production in the model. In this regard, why would eliminating a putative inhibitor of CXCL12 in the Gal-3 knockouts result in restoration of CXCR4-dependent activity after TG when CXCR4 has been fully blocked by the pharmacologic inhibitor of CXCR4 AMD? This is not consistent with the proposed role of Gal-3.”

Response: We agree with referee 2 that our findings may have “multiple interpretations”, and a contribution from other CC and CXC chemokines may be possible. We have now repeated the experiment to increase the number of mice studied, the new data add support to the contribution by the suggested mechanism: the difference in CXCR4-dependent migration with an increase in Gal-3 knockout mice as compared to wild-type mice is now significant.

e. “In the related Figure EV5A, the authors claim that TG induced recruitment of cells to the peritoneum that are PLA proximity assay positive. However, at most 10% of the cells are positive, and the cells are not identified. Do the PLA+ cells represent 50% of the classical monocytes? If not, how do the authors explain the results in panel B?”

Response:

As referee 2 correctly states, we did not identify the recruited leukocytes in the peritoneal lavage. The initial population used in Duolink and FACS experimental series was from the same peritoneal lavage from individual mice. We have now analyzed the percentage of leukocyte subsets using the FACS data and found that classical monocytes account for ~4% of the total cell population. In principle, all the Duolink-positive cells could be classical monocytes (Fig. S12E). Considering only the low number of PLA+ cells in original panel A (now panel C), it is possible that some or most of the PLA+ cells are indeed classical monocytes; however, we can definitely not exclude that there are other subtypes present. We assume that with Panel B, referee 2 was referring to the right panel of Figure 4A (now Expanded View Figure 4D). This figure has now been labeled

appropriately. Moreover, in this revised version, we now show additional representative micrographs.

New Expanded View Figure 4: continued

New Appendix Figure S12: Leukocyte subsets of the peritoneal lavage.

Point 6 -- Statement by Reviewer 2: “Figure S10I: there is no negative control. This is important in general, but more so in particular since the chemotactic index is very low in this system. Does 100 nM gal-1 also inhibit across this concentration range, for example?”

Response: As we discussed in our response to point 3 above, our presentation of the chemotactic index led to a misinterpretation by referee 2 due to lack of information that we previously had not provided. This has now been added to the manuscript. Furthermore, as requested by referee 2, we have added a new experiment (now Appendix Figure S11A, please see response to point 2) that shows the effect of Gal-1 over a CXCL12 concentration range. These results demonstrate that, in contrast to Gal-3 CRD, Gal-1 has no significant inhibitory effect even up to 100 nM.

3rd Editorial Decision

6 November 2019

Thank you for the submission of your further revised manuscript to our editorial offices. We have now received the report from the referee that was asked to re-assess your study, you will find below. As you will see, referee #2 has remaining concerns and suggestions to improve the manuscript I ask you to address in a final revised version, either by adding data or by text changes, and/or in a detailed point-by-point-response (in case you feel points have already been adequately addressed during the previous revision). Please provide a detailed point-by-point-response in any case.

Further, I have these editorial requests:

- Please move all the material and methods from the Appendix to the main manuscript text. All methods information should be in the main manuscript. Please remove then the additional methods from the Appendix.
- Please ensure that the grant information in the Acknowledgements of the manuscript text is similar to the one entered in the submission system. Please also check that both are complete.
- Please provide a shorter synopsis blurb (short 2-sentence summary of the paper). This should have not more than 35 words.

REFeree REPORT

Referee #2:

In this latest version of the paper, the flaws we highlighted in our previous review have not been satisfactorily addressed. In particular the weak functional data have been supplemented with additional weak chemotaxis experiments (low chemotaxis index, huge error bars). In addition, the paper remains very disorganized, jumping from appendices to expanded views to main figures with experiments all over the place and sometimes referring to controls that appear in separate panels (For example, in Figure 3 panels F and G don't include a WT control, you need to look at panels A and B). This makes the paper very hard to follow and not rigorously controlled. Together with the

weak functional effects, the evidence is just not convincing. I think the authors should do a better job in organizing the papers and consider bringing important experiments shown in the appendices into the main figures.

Specific comments:

1. Chemotaxis experiments. These remain a major weakness of the paper. The authors are working with very weak chemotaxis systems where at best they get 5,000 migrating cells with CXCL12 of the 100,000 cells they put on top of the filter (See their point-by-point responses to point 3). At this minimal level of chemotactic index, the specificity of any claim of inhibition has to be questioned. This problem of sensitivity of the assay system is only worsened by the large variance of the measurements. In some experiments only 150 of the 100,000 input cells migrate in response to chemokine without inhibitor. This is a poor assay from which to draw any meaningful conclusions.
2. The more convincing data that indeed Gal-3 might block CXCL12 are in appendix S11, particularly panel C and panels G, H, K, and L (I think these should be in the main figures). In my previous review these data were the reason I gave for considering accepting the paper, but certainly not in that format, with key data as a supplement.
3. In appendix S2, the authors say that heparin but not lactose interferes with the Gal-3: CXCL12 binding, but in the SPR experiment in panel B (CXCL12 + lactose) they show a R_{max} of only 4 units. This is too close to the detection limit of the machine and it is quite risky to make an affinity determination with such low binding. Can't the authors try to immobilize Gal-3 by thiol coupling instead of by biotinylation as they do with Gal-3 CRD, which apparently improves the binding recordings?
4. In appendix S8A and B. Despite the quantification they show on the right green bar panels with large error bars, I honestly cannot see a difference on the CXCL12 binding dot blots between the Gal-3 CRD mutants N160A, E185A, N222A, these three mutants bind to CXCL12 with same intensity based on the provided example dot blots. (E185A and N222A are supposed to be weaker CXCL12 binders than N160A and WT Gal-3 CRD). The absence of anti-CXCL12 activity of E185A and N222A is better illustrated in Figure 3F and 3G (although they do not compare in parallel with the WT in these panels). Figure Appendix S8 is quite confusing.
5. I do not think Figure Appendix S10 is the best way to prove that a Gal-3 CRD Q220K mutant is a poor CXCL12 binder. Any loading unbalance at the time of loading the gel may make the band of interest look dimmer, in fact, look at the irrelevant band right above 28 kDa, that band is also dimmer in the Q220K lane. I would suggest that the figure be removed.
6. In Appendix S9, the table in panel H is very confusing. For example to indicate 130 nM they write 13×10^1 , to indicate 1,500 nM they use 15×10^2 , and then other measurements are not indicated in scientific format. They should unify the way they write these affinities (use scientific format with the same power of 10 for all). Moreover, looking at the binding sensorgram insets of the panels A to G, it seems that an affinity determination by kinetics would have significantly change the KD that the authors show in panel H calculated on the equilibrium. For example, mutants H217A and Q220A have a much slower CXCL12 dissociation than the WT Gal-3, and therefore a much higher overall affinity, just the opposite of the results they show in panel H, which would argue against the binding mechanism model the author propose.
7. Figure 3H and 3I. How do the authors explain that N160A affects the inhibitory activity of Gal-3 differently when the mutation is performed on the full length protein (3H) than on the CRD (3I)?
8. In figure 4A, the authors show a complete abolition by Gal-3 of the CXCL12-induced recruitment of neutrophils to the peritoneal cavity. However, in vitro, in Figure appendix S11P the inhibition of Gal-3 of CXCL12-chemotaxis of neutrophils is minimal. How do the authors explain this?

Thank you for your letter of November 4, 2019 regarding our manuscript EMBOR-2019-47852V3 and the assessment from referee 2 on our previously revised version. By following your clear guidelines stating that you "...think these issues can be addressed mainly by re-arranging the paper, rewriting and moving some data (as indicated by the referee), but also by adding new data", I believe that we now have a much improved manuscript. In this regard, we have done substantial re-writing and re-arranging of parts of the paper, as well as adding new and message-strengthening data. Referee 2's thoughtfulness and diligence in reviewing our paper is highly appreciated and has helped us to improve the work. While the structural studies presented here were judged to be convincing, we realize that these functional studies are only the beginning.

In the previous versions of our manuscript, the title placed some emphasis on the functional aspect of the new type of pairing between chemokines and galectins. Given the referee's reasoning on this issue, we changed the title and mode of presentation. Our new title focuses on the novelty of the structural pairing between chemokines and galectins, and the data presented provide initial studies into their functional significance. Basically, we discovered a new type of molecular encounter between two well-studied (and popular) inflammatory mediators. We have addressed each of the reviewers concerns in our point-by-point responses. To this end, we hope that this final version of our paper has sufficiently addressed all of the remaining concerns of reviewer 2, and that our paper now meets your standards for acceptance.

Thank you for the submission of your further revised manuscript to our editorial offices. I have now received the report from the referee that was asked to re-assess your study, you will find below. As you will see, referee #2 has some remaining minor points and suggestions to improve the manuscript text, I ask you to address in the really final revised version. Please also provide a point-by-point-response addressing these points when submitting your revised paper.

Further, I have this final request:

- In the grant information in the Acknowledgements funding by the National Science Foundation and the University of Minnesota Medical School and the Minnesota Medical Foundation is mentioned. These grants have not been entered into the submission system. Please do that and check again that in the online form and the manuscript the funding information is the same and complete.

REFeree REPORT

Referee #2:

This reviewer appreciates the extensive effort made by the authors to answer my concerns. Since the first version, several new experiments have been added and this last version has clearly improved in clarity and organization. I recognize the originality and potential of this work. The authors convincingly demonstrate that galectins interact with chemokines but this interaction causes only a partial inhibition of the chemokine activity. In addition, the authors fall short at identifying the mechanism by which galectins may hinder the chemokine activity. The authors should consider the following changes before this paper is ready for publication in EMBO Reports.

1. It appears clear to this reviewer that galectins do not block the interaction of CXCL12 to its receptor CXCR4. This is directly supported by couple experiments in the paper: Fig. 6D, Gal3 CRD does not block CXCL12-mediated CXCR4 internalization; Fig. 6G, Gal3 CRD binds to the surface of lactose-treated cells only in complex with CXCL12, and this interaction is blocked in the presence of the CXCR4 antagonist AMD3100. And indirectly by the fact that galectins only achieve a partial inhibition of the chemokine functions (chemotaxis, cAMP, bArrestin). Leaving aside the quality of the chemotaxis experiments and the big experimental errors in most cases, averaging all the chemotaxis experiments presented in this paper, galectins reduce chemotaxis in approximately 58%, whereas, at the right dose, a chemokine-binding protein that competes the interaction of the ligand with the receptor should achieve 100% inhibition. In addition their model in Expanded Figure 5 shows that the binding of Gal3 CRD to CXCL12 allows the interaction of the chemokine to CXCR4. Therefore, changes in the text, especially in the discussion, should be included to unequivocally state that galectins don't block the chemokine-receptor interaction and rule out this as a possible mechanism for the galectin-mediated reduction of the chemokine activity. Of course, it is

possible that despite binding to CXCR4, the complex Gal-3CRD-CXCL12 may not be as potent as CXCL12 in triggering intracellular signals. This possibility may also be discussed in the Discussion.

2. The blockade of the chemokine binding to cell surface GAGs by galectins is a very likely mechanism for the galectin-mediated reduction of the chemokine functions. Consistent with this idea, the authors show in Figure 2 that galectins bind CXCL12 through basic residues known to be essential for the chemokine binding to GAGs. In addition, in appendix figure S1, they prove that soluble heparin compete the binding of CXCL12 to Gal3 CRD. Furthermore, the partial anti-chemokine effects of galectins resemble what others have found with soluble heparin *in vitro*. Intriguingly, Gal-3 seems to be a more potent anti-chemokine molecule when injected in the peritoneum (Figure 4), this may reflect the fact the GAG-binding is known to be more important for the chemokine activity *in vivo* than *in vitro*. A chemotaxis experiment in a GAG-deficient cell line would answer whether the anti-chemokine activity of galectins is GAG-dependent, but at the very least the authors should change the introduction and discussion to highlight the importance of GAGs in chemokine activity (currently very vaguely mentioned only in the discussion) and list this idea as a very possible mechanism for galectins.

OTHER MINOR COMMENTS:

3. Examples figure 1B and 1D show no interaction between Gal-3 and CXCL9 and CXCL11. However, summary figure 1E indicates that Gal-3 binds CXCL9 and CXCL11, does this mean that these interactions are not reproducible? If available, authors should include different example figures to represent the summary data more accurately, or correct summary data accordingly.

4. Page 5 line 10 reads "Immobilized CXCL12 bound Gal-3 CRD..." but it seems that in this experiment the immobilized component was Gal-3 CRD, so this sentence should read "Immobilized Gal-3 CRD bound CXCL12...". Please revise.

5. Fig. 4C is not very well described in the Results and it's hard to follow the authors' interpretation. This paragraph should be changed to help the reader understand the main messages of this figure: 1) Thioglycollate induces cell recruitment into the peritoneum; 2) Cell recruitment is partially reduced in the presence of the CXCR4-antagonist AMD3465, what indicates that a significant portion of the cells are recruited by CXCR4-independent mechanisms; 3) Cell recruitment is enhanced in the absence of Gal3, in both AMD3465-treated and non-treated animals, what may be explained by Gal-3 blocking other chemokines or inhibiting CXCL12 in a CXCR4-independent manner (once again, an observation that supports my comment number 1).

6. Page 13, line 8, where it says "Fig. 5H" should say Fig. 6H.

4th Revision - authors' response

16 January 2020

EMBOR-2019-47852V4

Point-by-point responses to referee #2:

This reviewer appreciates the extensive effort made by the authors to answer my concerns. Since the first version, several new experiments have been added and this last version has clearly improved in clarity and organization. I recognize the originality and potential of this work. The authors convincingly demonstrate that galectins interact with chemokines but this interaction causes only a partial inhibition of the chemokine activity. In addition, the authors fall short at identifying the mechanism by which galectins may hinder the chemokine activity. The authors should consider the following changes before this paper is ready for publication in EMBO Reports.

Point 1 as stated by Referee #2: "It appears clear to this reviewer that galectins do not block the interaction of CXCL12 to its receptor CXCR4. This is directly supported by couple experiments in the paper: Fig. 6D, Gal3 CRD does not block CXCL12-mediated CXCR4 internalization; Fig. 6G, Gal3 CRD binds to the surface of lactose-treated cells only in complex with CXCL12, and this interaction is blocked in the presence of the CXCR4 antagonist AMD3100. And indirectly by the fact that galectins only achieve a partial inhibition of the chemokine functions (chemotaxis, cAMP, bArrestin). Leaving aside the quality of the chemotaxis experiments and the big experimental errors in most cases, averaging all the chemotaxis experiments presented in this paper, galectins reduce chemotaxis in approximately 58%, whereas, at the right dose, a chemokine-binding protein that competes the interaction of the ligand with the receptor should achieve 100% inhibition. In addition their model in Expanded Figure 5 shows that the binding of Gal3 CRD to CXCL12 allows the interaction of the chemokine to CXCR4. Therefore, changes in the text, especially in the discussion, should be included to unequivocally state that galectins don't block the chemokine-receptor interaction and rule out this as a possible mechanism for the galectin-mediated reduction of the chemokine activity. Of course, it is possible that despite binding to CXCR4, the complex Gal-

3CRD-CXCL12 may not be as potent as CXCL12 in triggering intracellular signals. This possibility may also be discussed in the Discussion.”

Response: For us as authors, it is very rewarding and satisfying to read how – based on the evidence that we present in our manuscript – the reviewer builds a chain of corroborating data to conclude that chemokine-receptor interaction is possible while the galectin-3 (Gal-3) CRD is associated to CXCL12 so that a ternary complex forms: we agree with pleasure and therefore without hesitation and thus introduced the recommended change “to unequivocally state that galectins don’t block the chemokine-receptor interaction and rule out this as a possible mechanism for the galectin-mediated reduction of the chemokine activity”. That the reviewer then even provides a plausible mechanism for this to occur is not an optional addition to the Discussion but invaluable information for our readers: thank you!

Equally important, our data on heparin in binding assays and identification of contact sites on CXCL12 for Gal-3 prompted the reviewer to recommend a second text addition, leading us to the second point.

Point 2 as stated by Referee #2: “The blockade of the chemokine binding to cell surface GAGs by galectins is a very likely mechanism for the galectin-mediated reduction of the chemokine functions. Consistent with this idea, the authors show in Figure 2 that galectins bind CXCL12 through basic residues known to be essential for the chemokine binding to GAGs. In addition, in appendix figure S1, they prove that soluble heparin compete the binding of CXCL12 to Gal3 CRD. Furthermore, the partial anti-chemokine effects of galectins resemble what others have found with soluble heparin in vitro. Intriguingly, Gal-3 seems to be a more potent anti-chemokine molecule when injected in the peritoneum (Figure 4), this may reflect the fact the GAG-binding is known to be more important for the chemokine activity in vivo than in vitro. A chemotaxis experiment in a GAG-deficient cell line would answer whether the anti-chemokine activity of galectins is GAG-dependent, but at the very least the authors should change the introduction and discussion to highlight the importance of GAGs in chemokine activity (currently very vaguely mentioned only in the discussion) and list this idea as a very possible mechanism for galectins.”

Response: Bringing in glycosaminoglycans (GAGs) as physiological chemokine binding partners let a very tempting line of reasoning start, and, again, we read with pleasure that the reviewer proceeds from our experimental data (calling in vivo data presented in Figure 4 “intriguing”) to build an attractive hypothesis: as advised, we acted accordingly to “highlight the importance of GAGs in chemokine activity (currently very vaguely mentioned only in the discussion) and list this idea as a very possible mechanism for galectins”. Actually, the passage around references 52-55 in our revised Discussion is now particularly strengthened.

Looking at Gal-3 and heparin binding, Talaga et al. (Biochemistry 2016;55:4541-51) reported μM affinity for unmodified heparin and complete blocking of binding by lactose, suggestion “that GAGs primarily occupy the lactose/LacNAc binding site of Gal-3”. This observation adds strong support to the reviewer’s hypothesis so that we also added the (smart) suggestion to test a GAG-deficient cell line to our revised text. As a brief note in this special context, it will be necessary, considering the detection of molecular compensation of defects in glycosylation KO mice (an (at that time unexpected) upregulation of O-mannosylation in KO mice deficient for the three core 2 β 1,6-N-acetylglucosaminyltransferases (Ismail et al., Glycobiology 2011;21:82-98) or a polyLacNAc chain length increase upon decrease of N-glycan branching by N-acetylglucosaminyltransferase-2 (Mgat2) deficiency (H. Mkhikian et al, eLife 2016;5:e14814)) that maintain capacity to bind galectins. Equally intriguing and relevant, expression of glycosyltransferases for (re)tailoring terminal positions for lectin contacts (for selectins, siglecs or galectins) was found to be upregulated upon loss of the branching enzymes GnT-IVa/b “to preserve N-glycan branch complexity”, among them the sialyltransferases ST3Gal-IV and ST6Gal-I (Takamatsu et al., Glycobiology 2010;20:485-97; for discussion of perspectives, please see also Dam & Brewer, Glycobiology 2010;20:1061-4). It could thus be not impossible that knocking down local sulfation by inducing GAG deficiency may set a compensatory mechanism in motion (via sulfotransferases), requiring rigorous controls besides generation of the cell model itself.

Point 3 as stated by Referee #2: “Examples figure 1B and 1D show no interaction between Gal-3 and CXCL9 and CXCL11. However, summary figure 1E indicates that Gal-3 binds CXCL9 and CXCL11, does this mean that these interactions are not reproducible? If available, authors should include different example figures to represent the summary data more accurately, or correct summary data accordingly”

Response: As advised, we corrected the summarizing figure 1E and removed the respective blue fields for CXCL9 and CXCL11. The interaction of Gal-3 with CXCL9 and CXCL11 remained negative throughout all solid-phase experiments. As discussed in the point-by-point response of the first revision, the disparity with the positive interaction identified by SPR is due to the different methods.

Point 4 as stated by Referee #2. Page 5 line 10 reads "Immobilized CXCL12 bound Gal-3 CRD..." but it seems that in this experiment the immobilized component was Gal-3 CRD, so this sentence should read "Immobilized Gal-3 CRD bound CXCL12...". Please revise. "

Response: *done as advised*

Point 5 as stated by Referee #2. Fig. 4C is not very well described in the Results and it's hard to follow the authors' interpretation. This paragraph should be changed to help the reader understand the main messages of this figure: 1) Thioglycollate induces cell recruitment into the peritoneum; 2) Cell recruitment is partially reduced in the presence of the CXCR4-antagonist AMD3465, what indicates that a significant portion of the cells are recruited by CXCR4-independent mechanisms; 3) Cell recruitment is enhanced in the absence of Gal3, in both AMD3465-treated and non-treated animals, what may be explained by Gal-3 blocking other chemokines or inhibiting CXCL12 in a CXCR4-independent manner (once again, an observation that supports my comment number 1).

Point 6 as stated by Referee #2. Page 13, line 8, where it says "Fig. 5H" should say Fig. 6H.

Response: *done as advised*

And, as before, the list of the reviewer, in all points, gives us reason to be enormously grateful: our dialogue has been a phenomenal privilege and pleasure; having been guided to substantial improvements in all aspects of the manuscript is sincerely greatly appreciated. To summarize our feeling, we simply say: THANK YOU VERY MUCH!

Accepted

20 January 2020

I am very pleased to accept your manuscript for publication in the next available issue of EMBO reports. Thank you for your contribution to our journal.